# Direct serotonin release in humans shapes aversive learning and inhibition

**Michael J. Colwell** [1,2] ✉, **Hosana Tagomori** [1,2], **Fei Shang** [1,2], **Hoi Iao Cheng** [1,2], **Chloe E. Wigg** [1,2], **Michael Browning** [1,2], **Philip J. Cowen** [1,2], **Susannah E. Murphy** [1,2,3] & **Catherine J. Harmer** [1,2,3] ✉

The role of serotonin in human behaviour is informed by approaches which allow in vivo modification of synaptic serotonin. However, characterising the effects of increased serotonin signalling in human models of behaviour is challenging given the limitations of available experimental probes, notably selective serotonin reuptake inhibitors. Here we use a now-accessible approach to directly increase synaptic serotonin in humans (a selective serotonin releasing agent) and examine its influence on domains of behaviour historically considered core functions of serotonin. Computational techniques, including reinforcement learning and drift diffusion modelling, explain participant behaviour at baseline and after week-long intervention. Reinforcement learning models reveal that increasing synaptic serotonin reduces sensitivity for outcomes in aversive contexts. Furthermore, increasing synaptic serotonin enhances behavioural inhibition, and shifts bias towards impulse control during exposure to aversive emotional probes. These effects are seen in the context of overall improvements in memory for neutral verbal information. Our findings highlight the direct effects of increasing synaptic serotonin on human behaviour, underlining its role in guiding decision-making within aversive and more neutral contexts, and offering implications for longstanding theories of central serotonin function.

Understanding the function of central serotonin (or 5-hydroxytryptamine, 5-HT) has been a focal goal of neuroscience research for nearly a century[1], not least because of its central role in the effects of many psychiatric drugs, predominantly selective serotonin reuptake inhibitors (SSRIs) and street drugs (e.g., ±3,4-methylenedioxymethamphetamine [MDMA] and lysergic acid diethylamide)[2,3]. Serotonin is phylogenetically ancient, and its function translates across species to many lower- and higher-level behaviours; from feeding and sexual functioning to goal-directed, flexible cognition[4–7]. Amongst these, behavioural inhibition, memory, and aversive processing are historically considered the core, specialised functions of serotonin[8–11]. This is underpinned by converging preclinical and human work

involving in vivo manipulation of synaptic 5-HT, predominantly with SSRIs or depletion of its amino acid precursor tryptophan (TRP)[7,12], and observing behavioural change. In humans, however, marked differences in the direction of behavioural effects are observed across similar experimental approaches. For example, several studies report seemingly contradictory effects of SSRIs on tasks of aversive and reward processing (reinforcement learning); specifically, different reports show that SSRIs increase reward sensitivity[13], increase loss sensitivity and decrease reward sensitivity[14], and decrease sensitivity to both reinforcement valences[15]. Inconsistent behavioural effects of SSRIs are also observed across other domains, including behavioural inhibition and memory processing[12,16–22]; in some cases, these

[1]University Department of Psychiatry, University of Oxford, Warneford Hospital, Oxford, UK. [2]Oxford Health NHS Foundation Trust, Warneford Hospital, Oxford, UK. [3]These authors contributed equally: Susannah E. Murphy, Catherine J. Harmer. ✉e-mail: michael.colwell@psych.ox.ac.uk; catherine.harmer@psych.ox.ac.uk

behavioural changes align with those seen after tryptophan depletion (e.g., reduced cognitive flexibility) despite the expectation that they would have opposing effects on net synaptic 5-HT[16,23].

Determining a causal link between increased synaptic 5-HT and behaviour in humans via SSRIs is difficult due to the complex effects of SSRIs on 5-HT and colocalised neurotransmitter systems. For example, negative signalling feedback along the serotonergic pathway following autoreceptor activation early in treatment can limit cell firing, and therefore 5-HT release, in a regionally-specific manner[24-26]. Furthermore, deactivation of 5-HT transporters results in 5-HT uptake via dopamine transporters, leading to subsequent co-release of dopamine and 5-HT[27]. The effect of increased dopaminergic content and signalling is seen in acute and subchronic SSRI administration[28-32], observable in striatal, prefrontal, and hippocampal structures implicated in reward processing, behavioural inhibition and memory functioning[27,33-35].

Given the complex molecular and behavioural profile of SSRIs, alternative probes which increase synaptic 5-HT may help further clarify the role of 5-HT in human behaviour and cognition. One such alternative involves the use of a selective serotonin releasing agent (SSRA) (Fig. 1): unlike SSRIs which increase 5-HT levels indirectly through prolonging synaptic 5-HT, SSRAs stimulate direct exocytic release of 5-HT, without broad monoaminergic efflux (as seen in non-selective 5-HT releasers, such as MDMA)[36,37]. While SSRIs require ongoing neural firing for vesicular release of 5-HT into the synapse, the SSRA mechanism is not firing-dependent and thus not negated by dorsal raphe autoreceptor negative feedback which delays the therapeutic onset of action of SSRIs[38-40].

Until recently, it has been challenging to characterise the effects of SSRAs in humans because of the lack of available licensed pharmacological probes. However, in 2020, low dose fenfluramine (up to 26 mg daily; racemic mixture) was licensed for the treatment of Dravet epilepsy[41]. Preclinical work suggests low dose fenfluramine directly and rapidly increases synaptic 5-HT without modifying extracellular dopamine concentration in regions involved in mood regulation such as the striatum and hippocampus, in contrast to SSRIs[40,42-53]. Fenfluramine led to substantially greater extracellular 5-HT levels than the SSRI, fluoxetine, when administered at similar doses[54]. Further preclinical work suggests acute administration of fenfluramine increases synaptic 5-HT by 182-200% vs basal state[53,55], while subchronic administration (4-5 days) retains increases in net 5-HT without influencing 5-HT terminal structural integrity[53,56]. In humans, acute and subchronic administration of $d$-fenfluramine decreases serotonergic radioligand binding of [$^{18}$F] altanserin in a dose-dependent manner, suggestive of increased synaptic serotonin release in the brain[57,58]. With its recent relicensing for epilepsy syndromes[41], fenfluramine provides an opportunity to probe the neurobehavioural effects of SSRAs in humans to answer outstanding questions about the role of synaptic 5-HT in human behaviour.

Here, we use this now-accessible approach to directly increase synaptic 5-HT in humans, examining its influence on domains of behaviour historically considered core functions of serotonin: aversive processing, behavioural inhibition, and memory. In line with our hypothesis that fenfluramine would result in a pattern of behaviour opposite to that seen with tryptophan depletion[12,16,17,20,59-62], we show that increasing synaptic serotonin reduces reinforcement sensitivity

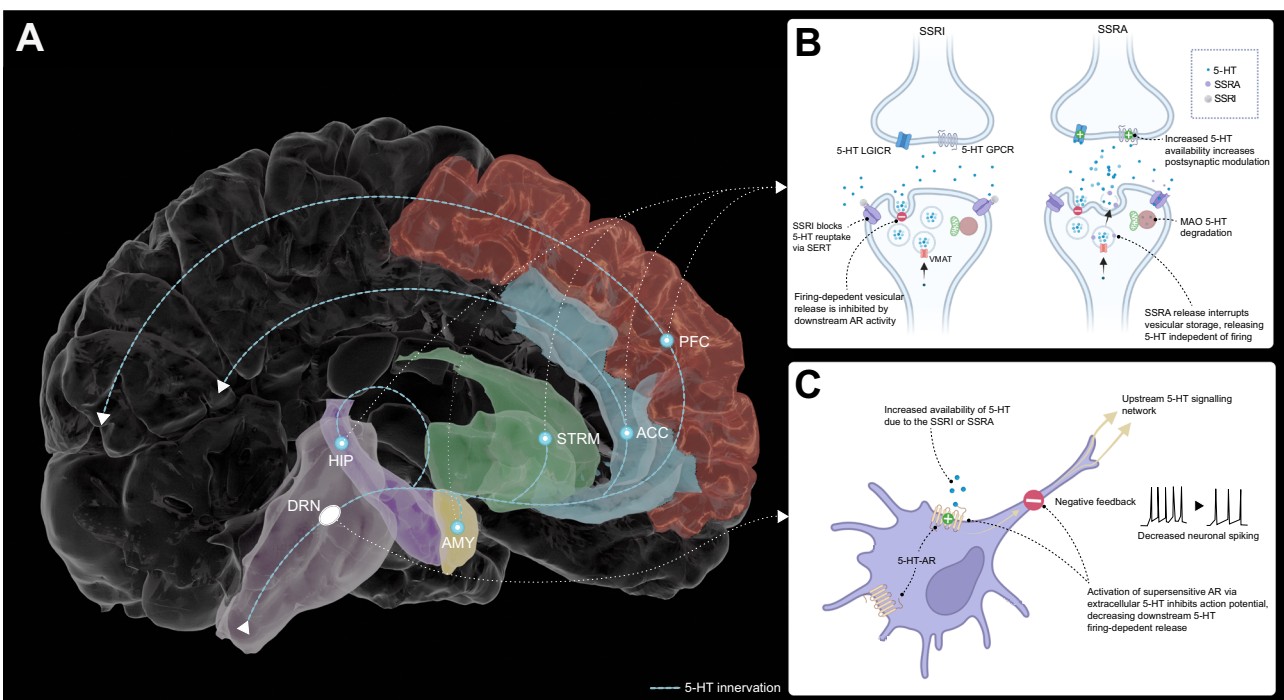

**Fig. 1 | Selective serotonin releasing agent is not negated by 5-HT$_{1A}$ supersensitivity, resulting in a rapid onset of pro-serotonergic activity. A** The majority of central serotonin (5-HT) innervation originates from the dorsal raphe nucleus (lilac), and is found within areas of the brain strongly implicated in mood regulation and cognitive function: amygdala (AMY; yellow), hippocampus (HIP; purple), striatal structures (STRM; green), anterior cingulate cortex (ACC; light blue) and the frontal lobe, including the prefrontal cortex (PFC; red). **B** Selective serotonin reuptake inhibitors (SSRI) and selective serotonin releasing agents (SSRA) both influence extracellular presynaptic serotonin concentrations, modulating downstream serotonergic activity, while the effects of SSRIs on synaptic 5-HT are delayed by autoreceptor hypersensitivity and may influence colocalised dopamine neurons. **C** 5-HT$_{1a}$ ARs are clustered in the dorsal raphe nucleus and are

endogenously sensitive to extracellular serotonin, and upon activation produce a negative feedback loop which inhibits upstream firing-dependent serotonin release. The original atlas meshes in panel **A** are credited to A. M. Winkler (Brain For Blender), which have been modified for illustrative purposes, released under a Creative Commons Attribution ShareAlike 3.0 Unported license (https://creativecommons.org/licenses/by-sa/3.0/). Panels B and C were created with BioRender.com and released under a Creative Commons Attribution-NonCommercial-NoDerivs 4.0 International license (https://creativecommons.org/licenses/by-nc-nd/4.0/deed.en). AR autoreceptor; GPCR G protein-coupled receptor; LGICR Ligand-gated ion-channel receptors; MAO Monoamine oxidase, SERT serotonin transporter.

for aversive outcomes; in addition, our findings indicate that increasing synaptic serotonin enhances behavioural inhibition and shifts bias towards impulse control during aversive interference. Finally, increasing synaptic serotonin increased memory for neutral verbal information.

## Results

### Does increased synaptic serotonin change reinforcement sensitivity for reward and loss?

We investigated the effect of increased synaptic 5-HT (via fenfluramine) on reinforcement sensitivity for reward and loss outcomes during a probabilistic instrumental learning task described in Fig. 2A[63,64]. During this task, participants learned the probability of outcomes associated with symbols within pairs. Each pair represented a task condition: win trials (win money or no change) and loss trials (lose money or no change). Within a given task block, each trial type was associated with one pair of symbols and appeared 30 times per block (across three blocks). Optimal choices were made when selecting symbols which had a greater probability (70%) of leading to a favourable outcome (i.e., win in win trials and no change in loss trials). A computational reinforcement learning model was fit to participant choice behaviour to formalise the predicted change in optimal choices between allocation groups. Model parameters for each trial type were derived, providing a distinct explanation of learning and decision-making behaviour throughout the task: learning rate ($\alpha$), explaining the rate at which outcomes modify expectations, which was estimated separately for win and loss trials; and outcome sensitivity ($\rho$), explaining the effective magnitude of experienced outcomes. Further information about the model, including parameter recovery and simulation procedures, are detailed within the Supplementary Methods.

In line with our hypothesis, fenfluramine allocation reduced the number of optimal choices during loss trials (ANCOVA group x task condition: F[1,50] = 5.14, $p = 0.03$, $\eta_p^2 = 0.07$ [95% CI 0.00, 0.24]; reward condition EMM = 0.68 ± 3.18, $p = 0.83$; loss condition EMM ± SE = −8.62 ± 3.18, $p < 0.01$, Cohen's $d = -0.75$ [95% CI −1.30, −0.19]) (Fig. 2B-C). Consistent with this, learning models fit to the data revealed increased synaptic 5-HT reduced outcome sensitivity for loss trials (ANCOVA group x task condition: F[1,50] = 5.73, $p = 0.02$, $\eta_p^2 = 0.10$ [0.00, 0.28]; reward condition EMM = 0.10 ± 0.43, $p = 0.82$; loss condition EMM = −0.90 ± 0.43, $p = 0.04$, $d = -0.57$ [−1.11, −0.03]) (Fig. 2D). In contrast, modelled learning rate for both conditions did not vary across groups in a statistically significant manner (ANCOVA group x task condition: F[1,50] = 1.22, $p = 0.27$; main effect of group: F[1,50] = 0.92, $p = 0.34$) (Supplementary Fig. 10). Similarly, fenfluramine allocation increased time to choice selection during loss conditions (ANCOVA group x task condition: F[1,50] = 5.52, $p = 0.02$, $\eta_p^2 = 0.11$ [0.00, 0.29]; reward condition EMM = 13.9 ± 95.6, $p = 0.89$; loss condition EMM = 246.0 ± 95.6, $p = 0.01$, $d = 0.71$ [0.15, 1.26]) (Fig. 2F), which would also be consistent with a relative reduction in loss sensitivity in this group.

Overall, these findings demonstrate that net increases in synaptic 5-HT (via serotonin releasing agent fenfluramine) decrease reinforcement sensitivity to loss outcomes, opposite to the effect of 5-HT depletion (TRP) where loss sensitivity increases[61,62]. While alternative computational accounts for the observed behaviour could include increased value decay or choice stochasticity, there are no reports of 5-HT manipulation influencing these components of behaviour[64].

### Does elevated 5-HT modulate behavioural inhibition, choice impulsivity, and vulnerability to aversive emotional interference?

Next, we assessed the impact of increased synaptic 5-HT on response inhibition (an index of behavioural inhibition), choice impulsivity, and interference during the Affective Interference Go/No-Go task. In this task, participants respond (go) or withhold responses (no-go) according to rules which change over time (e.g., instructions: do not press the button if you see a blue/yellow image) while being exposed to emotional distractors (fearful or happy faces, or control images) (Fig. 3A). Fenfluramine allocation increased response inhibition, measured by mean percentage of accurately withheld responses to no-go trials (ANCOVA main effect of group: F[1,47] = 11.26, $p < 0.01$, $\eta_p^2 = 0.15$ [0.00, 0.37]; all conditions EMM = 9.69 ± 2.63, $p < 0.001$, $d = 0.60$ [0.27, 0.93]) (Fig. 3B). Further, there was no statistically significant group effect for go trial accuracy (ANCOVA main effect of group: F[1,47] = 0.83, $p = 0.37$) (Supplementary Fig. 14B).

Signal detection theory analyses was undertaken to determine if group differences in response inhibition were driven by perceptual decision-making (Fig. 3C). Fenfluramine allocation resulted in more cautious decision-making throughout (log criterion $c$; ANCOVA main effect of group: F[1,47] = 13.54, $p < 0.001$, $\eta_p^2 = 0.19$ [0.02, 0.39]; all conditions EMM = 0.08 ± 0.02, $p < 0.001$, $d = 0.39$ [0.16, 0.62]) (Fig. 3D), but there was no statistically significant group effect on signal discriminability (see Supplementary Note 4).

There was a positive relationship between go trial response times and both response inhibition ($r = 0.61$, $p < 0.001$, two-tailed) and cautious decision-making (log decision criterion [$c$]; $r = 0.77$, $p < 0.001$, two-tailed), as shown in Supplementary Fig. 15. These findings represent an inherent speed-accuracy trade-off within the limited response window for the task (400 ms), where a favourable optimisation for task strategy is slowing response times to ensure greater response inhibition accuracy. As a result, faster response time for go trials may be considered an index of choice impulsivity. Indeed, it is argued that the tendency to react in a premature manner without adequate signal processing constitutes the fundamental aspects of impulsive behaviour[65,66].

Increasing synaptic 5-HT (via fenfluramine) resulted in reductions in choice impulsivity, indicated by increased time to choice for go trials, across all task conditions (ANCOVA main effect of group: F[1,47] = 22.00, $p < 0.001$; $\eta_p^2 = 0.27$ [0.07, 0.46]) (Fig. 4). Moreover, there was an interaction between group and task interference (happy, fearful or control distractors) on choice impulsivity (ANCOVA group x task condition: F[2,95] = 3.22, $p < 0.05$, $\eta_p^2 = 0.08$ [0.00, 0.20]). Specifically, choice impulsivity was most reduced when aversive emotional distractors were present in the fenfluramine group (EMM = 21.3 ± 4.71, $p < 0.0001$, $d = 1.28$ [0.70, 1.86]) compared with both control (EMM = 14.6 ± 4.71, $p < 0.01$, $d = 0.88$ [0.31, 1.44]) and positive emotional distractors (EMM = 15.4 ± 4.71, $p < 0.01$, $d = 0.93$ [0.36, 1.50]). In a separate analysis of the fenfluramine group, there was a main effect for valence on choice impulsivity when isolating aversive vs control conditions (ANCOVA main effect: F[1,22] = 4.87, $p = 0.04$, $\eta_p^2 < 0.01$ [0.00, 0.19]); however, there was no statistically significant main effect of valence in the placebo group (see Supplementary Table 8).

Computational drift diffusion modelling (Fig. 5A) was undertaken to investigate evidence accumulation patterns throughout the Affective Interference Go/No-Go task (for further model details, including recovery and simulation procedures, see Supplementary Methods). Fenfluramine allocation shifted initial choice bias ($z \cdot a$) toward impulse control (no-go, lower boundary) during aversive interference only (ANCOVA group x task condition: F[2,95] = 3.45, $p = 0.03$, $\eta_p^2 = 0.06$ [0.00, 0.17]; control condition EMM = −0.01 ± 0.15, $p = 0.96$; positive interference EMM = −0.16 ± 0.15, $p = 0.31$; aversive interference EMM = −0.33 ± 0.15, $p = 0.03$, $d = -0.60$ [−1.17, −0.04]) (Fig. 5B). There was no statistically significant group effect across other model parameters, including boundary separation ($a$) and drift rate ($v$) (see Supplementary Note 4). As 75% of task trials fit

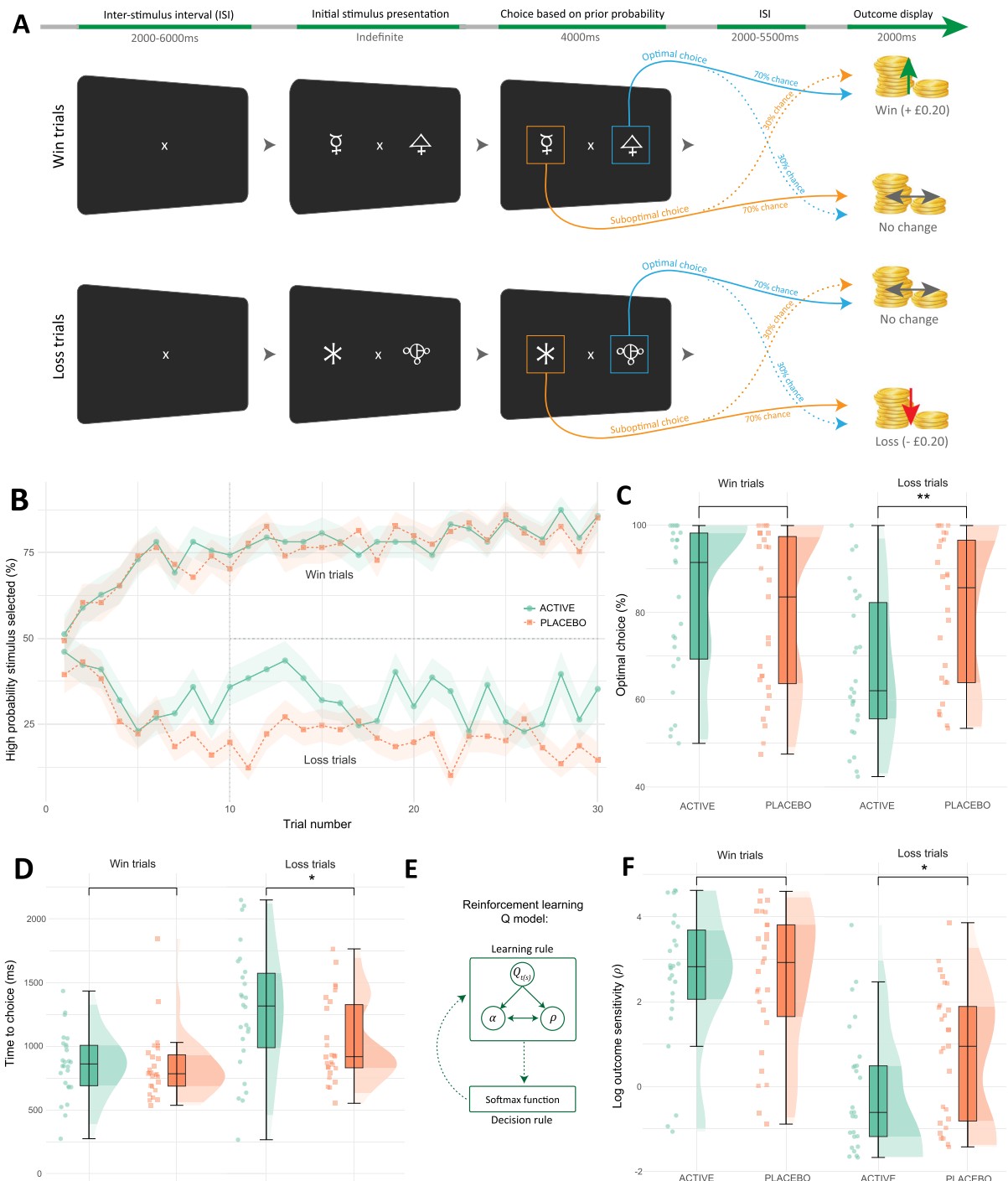

**Fig. 2 | Task procedure, computational modelling, and analyses of the probabilistic instrumental learning task. A** The task consists of choosing a symbol within a pair, with two interleaved pairs per task block. Each pair represents a high probability of winning (win trials) or high probability of loss (loss trials). Win trials (30 per block) result in winning or no change, while loss trials (30 per block) result in loss or no change. Within pairs, symbols had fixed reciprocal probabilities (70%, 30%), with outcomes displayed at trial end. Participants were instructed to make choices for most likely maximal monetary gain (awarded at study completion). **B** Group learning rates across trial types (blocks averaged). High probability stimulus selected (Y axis): mean percentage of high probability win or loss choices. The shaded area around lines represents standard error (SE). **C** Decreased optimal choices in the fenfluramine (active) group during loss trials via estimated margin means (EMM) (reward trials EMM($\pm$ SE) = 0.68 $\pm$ 3.18, $p$ = 0.8307; loss trials EMM = $-8.62 \pm 3.18$, $p$ = 0.0078, Cohen's $d$ = $-0.75$ [95% CI $-1.30$, $-0.19$]).

**D** Increased response time (ms; milliseconds) in the fenfluramine group during loss trials (reward trials EMM = 13.9 $\pm$ 95.6, $p$ = 0.8845; loss trials EMM = 246.0 $\pm$ 95.6, $p$ = 0.0115, $d$ = 0.71 [0.15, 1.26]). **E** The Q computational model contains two primary parts: a learning rule and decision rule. The learning rule describes trial-by-trial updates of value expectation ($Q_{t(s)}$), and choice probability is determined via the decision rule. Model parameters alter distinct aspects of the decision-making process: outcome sensitivity ($\rho$) and learning rate ($\alpha$) (see Supplementary Methods for details)[64]. **F** Decreased outcome sensitivity ($\rho$) in the fenfluramine group during loss trials (reward trials EMM = 0.10 $\pm$ 0.43, $p$ = 0.8203; loss trials EMM = $-0.90 \pm$ 0.43, $p$ = 0.0392, $d$ = $-0.57$ [$-1.11$, $-0.03$]). **B**–**D**, **F** include $N$ = 53 individuals; box-plots represent interquartile range (IQR); central line depicts the median. Whiskers represent $\pm$1.5 IQR, encompassing most data points; half-violin plots depict the data distribution. ** $p \le 0.01$, * $p \le 0.05$ indicate group differences by two-tailed EMM tests (Bonferroni-Holm corrected). ISI Inter-stimulus interval.

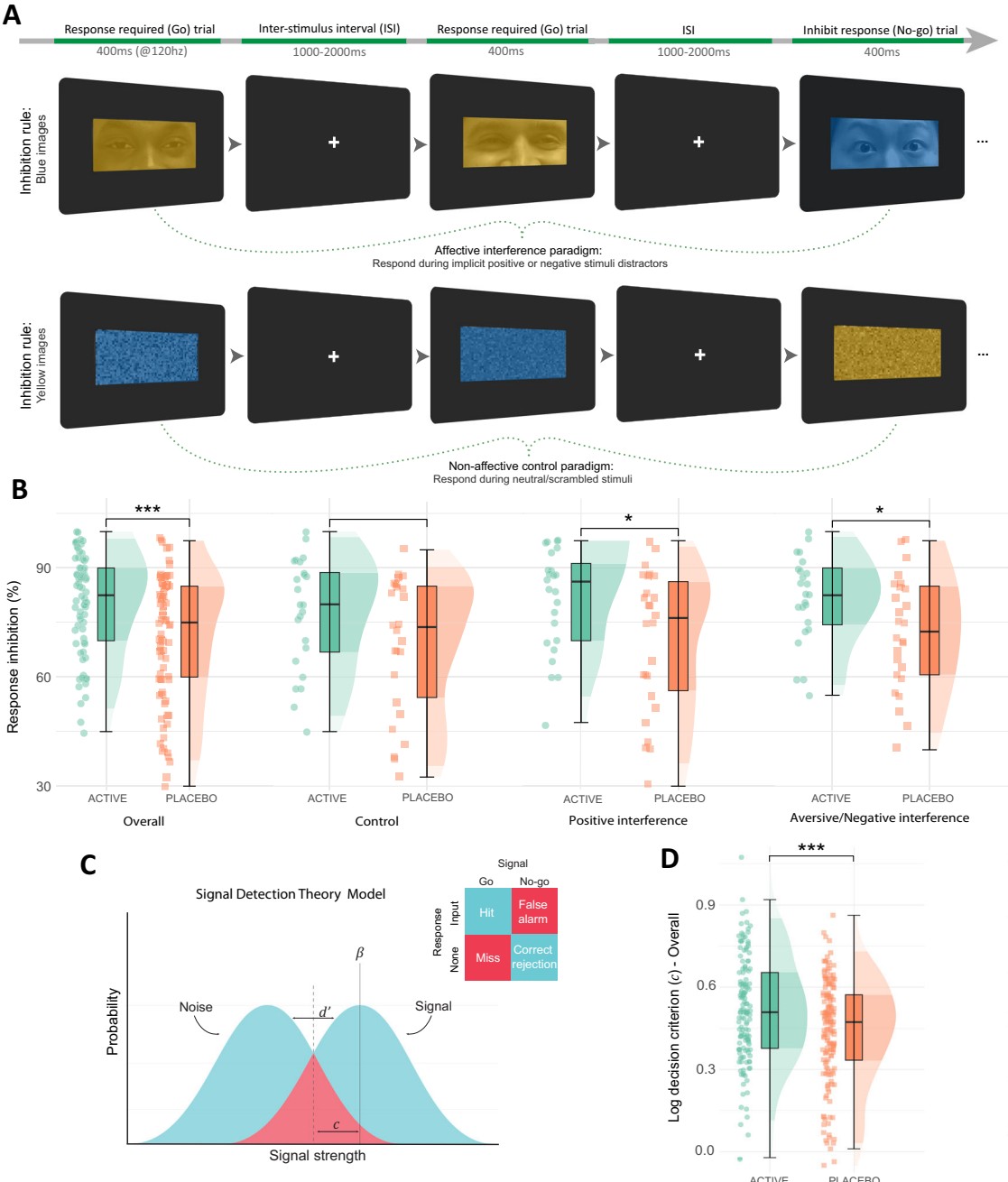

**Fig. 3 | Task procedure and accuracy analyses for the Affective Interference Go/No-Go task. A** An example of trial flow across two blocks from the affective interference go/no-go task, with one block depicting affective interference (above) and the other showing the non-affective (scrambled) control paradigm (below)[143]. The sequence of trials is left to right. The first two trials in each example illustrate go trials where participants respond with a key input (80% of trials); the third trial in the sequence illustrates a no-go trial where participants must withhold responses (20% of trials). There were six task blocks (two per task condition), with 80 trials within each block. Partial faces displayed in the top row are from the RADIATE stimulus set[144,145]. Models in each image consented to the photography and release of their photos for research purposes. **B** Higher response inhibition (mean %) performance was observed in the fenfluramine (active) group compared with the placebo group across all conditions via estimated marginal means tests (EMM) (overall EMM = 9.69 ± 2.63, $p = 0.0003$, $d = 0.60$ [0.27, 0.93]; control EMM = 8.58

± 4.56, $p = 0.0616$; positive interference EMM = 11.25 ± 4.56, $p = 0.0147$, $d = 0.69$ [0.13, 1.26]; aversive interference EMM = 9.25 ± 4.56, $p = 0.0442$, $d = 0.58$ [0.01, 1.14]). **C** Application of signal detection theory indices to the go/no-go task, where correct and incorrect go/no-go responses are described on a sensory continuum of noise and signal (see further details in Supplementary Methods). **D** Fenfluramine allocation resulted in higher values for signal detection theory criterion index $c$ (indicative of more conservative/cautious decision-making) across all task conditions (EMM = 0.08 ± 0.02, $p = 0.0007$, $d = 0.39$ [0.16, 0.62]). **B**, **D** Include data for $N = 50$ individuals; boxplots represent the interquartile range (IQR), while the central line depicts the median. The whiskers extend to approximately ± 1.5 times the IQR, encompassing the bulk of the data points; half-violin plots depict the data distribution; ***$p \leq 0.001$, **$p \leq 0.01$, *$p \leq 0.05$ indicate group differences by two-tailed EMM tests (Bonferroni-Holm corrected). ISI Inter-stimulus interval.

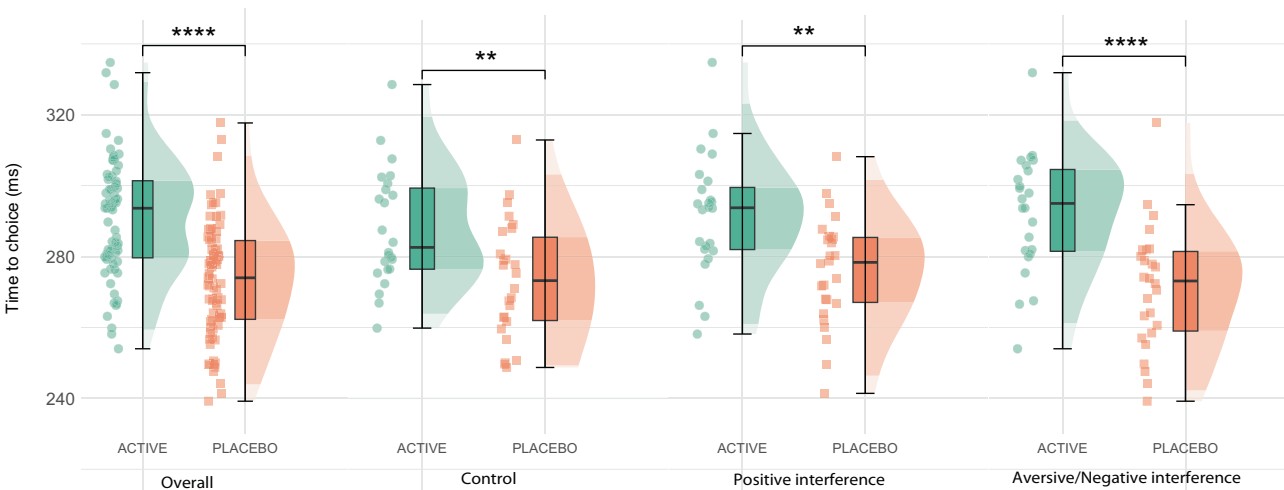

**Fig. 4 | Time to choice analysis for the Affective Go/No-Go Task.** General decreases in choice impulsivity (or, choice time for correct go trials) (ms; milliseconds) were observed in the fenfluramine (active) group via estimated marginal means tests (EMM) (overall EMM = 17.2 ± 2.72, $p$ = 3.733661e-09, $d$ = 1.03 [0.68, 1.37]; control EMM = 14.6 ± 4.71, $p$ = 0.0024, $d$ = 0.88 [0.31, 1.44]; positive interference EMM = 15.4 ± 4.71, $p$ = 0.0013, $d$ = 0.93 [0.36, 1.50]; aversive interference EMM = 21.3 ± 4.71, $p$ = 1.31162e-5, $d$ = 1.28 [0.70, 1.86]); this effect was most

pronounced during aversive interference. Figure includes data for $N$ = 50 individuals; boxplots represent the interquartile range (IQR), while the central line depicts the median. The whiskers extend to approximately ± 1.5 times the IQR, encompassing the bulk of the data points; half-violin plots depict the data distribution; **** $p \leq 0.0001$, ** $p \leq 0.01$ indicate group differences by two-tailed EMM tests (Bonferroni-Holm corrected).

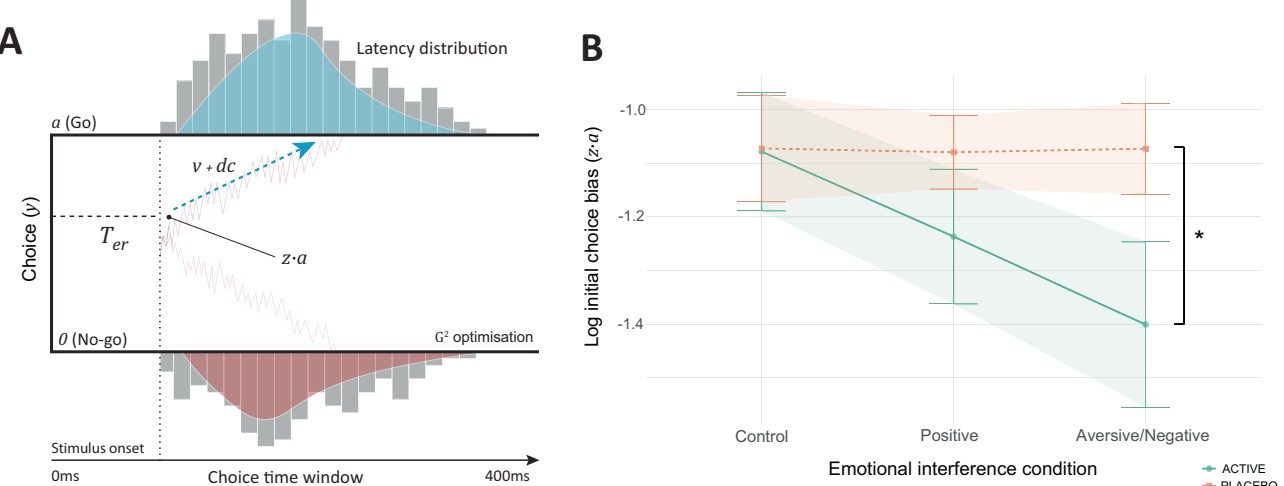

**Fig. 5 | Computational drift diffusion modelling and choice bias during affective interference. A** A drift diffusion model (DDM) was fit to participant behaviour during the Affective Go/No-Go Task. The DDM provides distinct explanations for evidence accumulation before reaching a decision (i.e., Go or No-Go) during the task, explaining distinct parts of the decision-making process which contribute to task behaviour. The DDM describes decision-making using five parameters: 1) Boundary separation ($a$), which describes the required quantity of evidence for making a decision. 2) Non-decision time ($T_{er}$) is the period between stimulus onset and the start of the evidence accumulation, where foremost sensory and perceptual processes occur, notably emotional facial expression encoding[146]. 3) Initial choice bias ($z \cdot a$) represents bias toward one of the choice boundaries ($a$ [Go] and 0 [No-go]) at the start of evidence accumulation. 4) Drift rate ($v$) describes the rate of evidence accumulation before arriving at a choice boundary. 5) Drift criterion ($dc$) is a constant applied to the mean drift rate which is evidence independent. The model was fit to behaviour using the Gsquare ($G^2$) approach which uses maximum

likelihood estimation, where choice time distributions were divided into five quantiles: 10th, 30th, 50th, 70th and 90th[147,148]. Model-fitted synthetic data and observed task data were closely matched, and all model parameters were recoverable (for further details, please see Supplementary Methods). **B** During interference with aversive emotional information (fearful faces), fenfluramine (active) allocation resulted in an initial choice bias ($z \cdot a$) toward the impulse control (no-go) choice boundary via estimated marginal means tests (EMM) (control condition EMM = −0.01 ± 0.15, $p$ = 0.9692; positive interference EMM = −0.16 ± 0.15, $p$ = 0.3093; aversive interference EMM = −0.33 ± 0.15, $p$ = 0.0352, $d$ = −0.60 [−1.17, −0.04]) ($N$ = 50). This suggests group differences in go trial choice time observed during the task, specifically during aversive interference, were driven by a bias toward the no-go boundary in the fenfluramine group. **B** Includes data for $N$ = 50 individuals; lines and plot points depict mean value, with error bars and shaded areas around each line depicting standard mean error; * $p \leq 0.05$ indicates group difference by two-tailed EMM tests (Bonferroni-Holm corrected).

to the DDM were go trials, and there was no statistically significant group effect on accuracy for these trials, group differences in model parameters may not occur when accuracy is similar despite differences in choice time[67].

Taken together, these findings suggest that increasing synaptic 5-HT enhances behavioural inhibition across emotional and more neutral contexts, an effect which was driven by more cautious decision-making. Moreover, increased 5-HT levels appear to shift bias towards

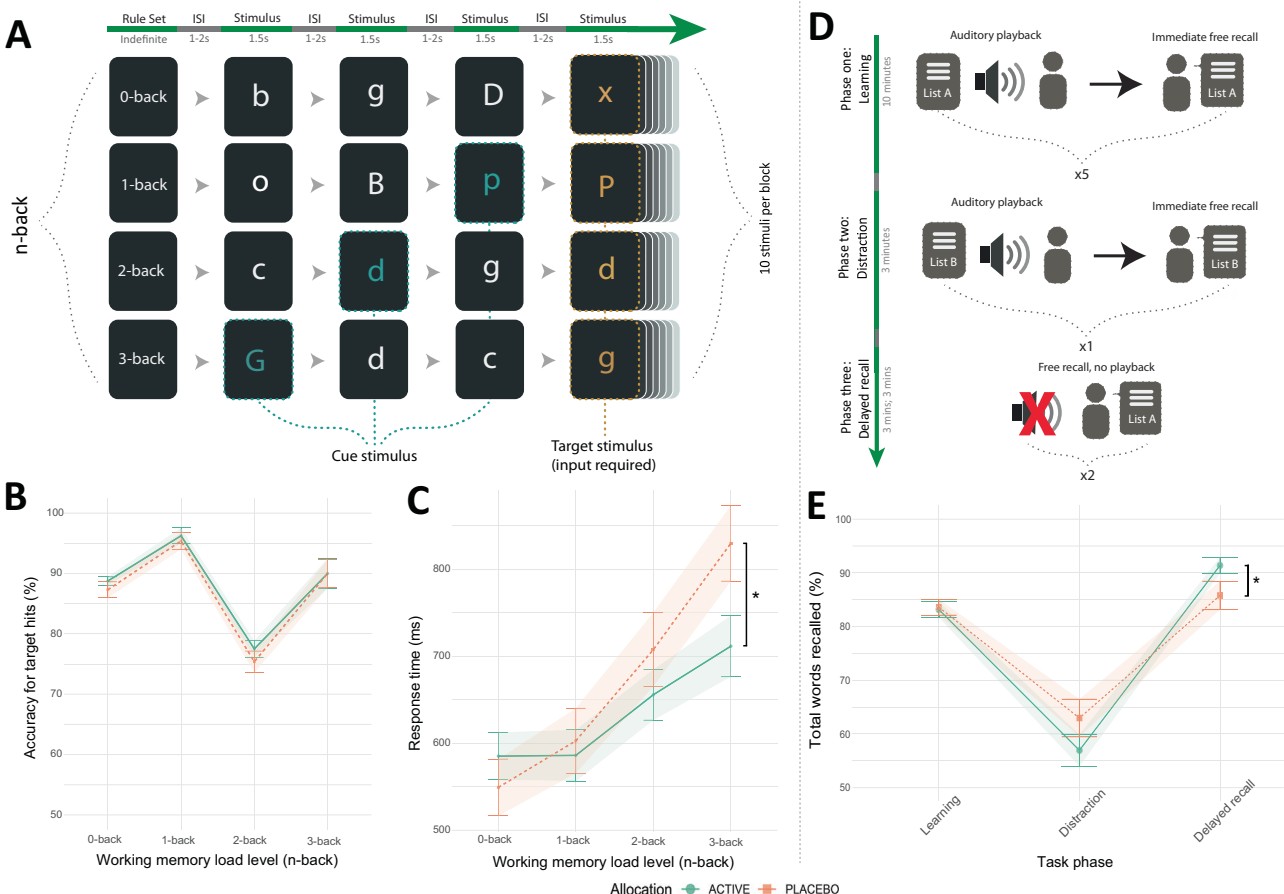

**Fig. 6 | Effects of the serotonin releasing agent fenfluramine across tasks of memory function (n-Back and Auditory Verbal Learning Task). A** Verbal n-back task example task flow for all four task conditions (top to bottom: 0-back, 1-back, 2-back and 3-back). The sequence of trials is left to right. Before each block of 10 stimuli, participants were given a rule for targets (e.g., instructions: press spacebar if you see the same letter that appeared two letters ago [2-back]). Each condition was repeated four times (16 blocks total). **B** No statistically significant difference in target accuracy was observed across groups (Supplementary Note 6). **C** Reduced response time (ms; milliseconds) for correct choices in the fenfluramine (active) group at highest n-back memory load via estimated marginal means tests (EMM) (0-back EMM: 35.9 ± 50.20, *p* = 0.4760; 1-back EMM: −16.7 ± 50.20, *p* = 0.7395; 2-back EMM: −51.8 ± 50.20, *p* = 0.3035; 3-back EMM: −118.20 ± 50.20, *p* = 0.0196, d = −0.67 [−1.23, −0.10]). **D** Auditory Verbal Learning Task flow across three task phases: phase one (learning/encoding), phase two (distraction), and phase three (delayed recall).

During phase one, participants listened to a recording of 15 verbal items (List A) at a slowed pace (1 s interval between words), followed by an immediate free recall of list items. After this occurred five times, phase two (distraction) required learning a novel list of items for immediate recall (List B). Phase three (delayed recall) required free recall (without list playback) of items from List A immediately after phase two and then fifteen minutes later. **E** The fenfluramine group showed increased accuracy during the delayed recall phase of the Auditory Verbal Learning Task relative to placebo (learning EMM = −0.07 ± 0.14, *p* = 0.6378; distraction recall EMM = −0.90 ± 0.70, *p* = 0.1961; delayed recall EMM = 0.84 ± 0.35, *p* = 0.0165, d = 0.34 [0.06, 0.61]). **B**, **C** contain data for *N* = 50 individuals, while (**E**) contains data for *N* = 51 individuals; lines and plot points depict mean value, with error bars and shaded areas around each line depicting standard mean error; *$p \leq 0.05$ indicate group differences by two-tailed EMM tests (Bonferroni-Holm corrected). ISI Inter-stimulus interval.

---

impulse control during aversive affective interference at the start of evidence accumulation, consequentially lowering choice impulsivity.

### Assessing the influence of increased synaptic serotonin on memory processing

Finally, we assessed the influence of fenfluramine administration on memory function. During a task of complex verbal working memory processing (Verbal n-back; Fig. 6A), participants were required to recall if a target letter occurred within a pre-specified sequential pattern (i.e., 0-, 1-, 2-, or 3-back letters ago). There was no statistically significant group effect for total number of correctly recalled targets (ANCOVA group analysis: F[1,47] = 0.70, *p* = 0.41) (Fig. 6B). However, during the highest task difficulty (3-back), fenfluramine allocation resulted in faster recall for correct targets (ANCOVA group x task condition: F[3,143] = 3.66, *p* = 0.01, $\eta_p^2$ = 0.05 [0.00, 0.13]; 0-back EMM: 35.9 ± 50.20, *p* = 0.48; 1-back EMM: −16.7 ± 50.20, *p* = 0.74; 2-back EMM: −51.8 ± 50.20, *p* = 0.30; 3-back EMM: −118.20 ± 50.20, *p* = 0.02, *d* = −0.67 [−1.23, −0.10]) (Fig. 6C).

During a task of long-term memory encoding and retrieval (Auditory Verbal Learning Task; Fig. 6D), participants were required to learn a list of 15 verbal items and correctly recall these items during learning (immediate recall) and after a short period (delayed recall). Fenfluramine allocation resulted in higher total accuracy during delayed recall, while there was no statistically significant group effect for learning trials (ANCOVA group x task condition: F[2,1474] = 6.23, *p* = 0.01, $\eta_p^2$ < 0.01 [0.00, 0.02]; delayed recall EMM = 0.84 ± 0.35, *p* = 0.02, *d* = 0.34 [0.06, 0.61]; learning EMM = −0.07 ± 0.14, *p* = 0.63; distraction recall EMM = −0.90 ± 0.70, *p* = 0.20) (Fig. 6E). There was no statistically significant group effect for word repetitions or intrusions (Supplementary Fig. 19).

There was no statistically significant group effect for performance on tasks of visuo-spatial working memory (Oxford Memory Task) and implicit visual learning (Contextual Cueing Task) (see Supplementary Note 7–8, Supplementary Table 4 and Supplementary Figs. 22, 24).

Taken as a whole, these findings suggest increasing synaptic 5-HT enhances memory processing for verbal information.

**Table 1 | Subjective outcome measures of cognition, affect, and mood across allocation groups – post-intervention descriptive statistics and inferential analysis**

| | Fenfluramine (n = 26) M (S.D.) | Placebo (n = 27) M (S.D.) | Inferential analysis [a] | |
| --- | --- | --- | --- | --- |
| | | | F-statistic [df] | p |
| Cognition | | | | |
| Perceived Deficits Questionnaire | | | | |
| Baseline | 10.27 (11.80) | 15.12 (13.60) | -- | -- |
| Follow-up | 13.46 (13.49) | 13.35 (12.99) | 0.01 [1,37] | 0.781 |
| Affect | | | | |
| Positive and Negative Affect Schedule | | | | |
| Negative items, baseline | 10.86 (1.28) | 12.00 (1.81) | -- | -- |
| Negative items, follow-up | 10.86 (1.24) | 11.61 (1.97) | 5.00 [1,38] | 0.0313 [b] |
| Positive items, baseline | 30.09 (6.37) | 28.17 (7.76) | -- | -- |
| Positive items, follow-up | 27.62 (6.74) | 26.22 (7.90) | 0.277 [1,38] | 0.602 |
| Visual Analogue Scale | | | | |
| Negative items, baseline | 148.92 (87.76) | 171.55 (68.73) | -- | -- |
| Negative items, follow-up | 152.12 (86.90) | 155.62 (79.94) | 0.29 [1,46] | 0.593 |
| Positive items, baseline | 736.13 (110.33) | 664.38 (83.23) | -- | -- |
| Positive items, follow-up | 699.62 (101.57) | 710.15 (99.07) | 0.00 [1,44] | 0.984 |
| Mood | | | | |
| Beck Depression Inventory | | | | |
| Baseline | 2.92 (3.14) | 5.00 (4.44) | -- | -- |
| Follow-up | 3.08 (2.99) | 2.60 (3.24) | 0.05 [1,43] | 0.826 |
| Spielberger Trait Anxiety Subscale | | | | |
| Baseline | 23.32 (5.51) | 23.09 (3.99) | -- | -- |
| Follow-up | 23.00 (5.64) | 22.38 (5.03) | 0.07 [1,36] | 0.790 |
| Spielberger State Anxiety Subscale | | | | |
| Baseline | 1.52 (2.00) | 2.26 (2.47) | -- | -- |
| Follow-up | 1.48 (1.83) | 2.33 (3.00) | 3.03 [1,36] | 0.090 |

[a]Inferential analysis via baseline-adjusted ANCOVA modelling across allocation groups (active vs placebo).
[b]Post-hoc EMM analysis (two-tailed) revealed no significant score difference on negative PANAS items across allocation groups (EMM = −0.75 ± 0.50, 95% CI [−1.76, 0.26], p = 0.141), while at baseline placebo scored significantly higher on negative PANAS items than fenfluramine (EMM = −1.14 ± 0.46, 95% CI [−2.06, −0.21], p = 0.02, Hedges' g = −0.70). Between visits, the mean score for this item did not change in fenfluramine group while the placebo group reduced by a mean difference of 0.39.

### Relationship of elevated 5-HT with self-report questionnaire measures, gender-related covariance, and cortisol concentration

There was no statistically significant group effect on self-report ratings of subjective cognition, side effects, motivation, and affect (Tables 1–2; Supplementary Table 3). At study completion, the placebo group (70%) were better than chance at correctly guessing their allocation compared with the fenfluramine group (50%), however this was not a statistically significant difference ($\chi^2$ = 3.92, p = 0.27). Gender did not covary with the effects of fenfluramine administration on task behaviour reported in previous sections (Supplementary Table 5). There was no statistically significant difference between groups in salivary cortisol during the initial dosing period, suggesting a lack of acute modulation of hypothalamic-pituitary-adrenal (HPA) axis function from fenfluramine.

## Discussion

The present findings demonstrate the direct effects of increased synaptic serotonin on human behaviour, underlining its role in guiding decision-making across aversive and more neutral contexts (i.e., where valence is not explicitly manipulated). Specifically, we observed reduced sensitivity for outcomes in aversive contexts; enhanced behavioural inhibition and increased bias favouring impulse control during aversive interference; and enhanced memory function for verbally encoded information. These findings offer broad implications for longstanding theories of how central 5-HT influences human behaviour and contributes to psychiatric aetiology.

Throughout instrumental learning, increased synaptic 5-HT (via fenfluramine) reduced sensitivity to aversive outcomes. This effect is opposite to that described following central depletion of serotonin with tryptophan-depletion, where enhanced negative prediction errors during probabilistic instrumental learning and bias toward aversive stimuli during Pavlovian conditioning have been observed[8,60,62,68]. Further, in a Pavlovian-to-instrumental transfer paradigm, independent depletion of 5-HT and dopamine respectively enhanced aversive and decreased rewarding Pavlovian-to-instrumental transfer[69]. As SSRAs and TRP result in opposite effects on net synaptic 5-HT, the opposite behavioural pattern observed here is consistent with a key role for serotonin in modulating loss sensitivity[12,38].

There was no statistically significant difference in reward processing between the SSRA and placebo groups. Despite the shared purpose of increasing synaptic 5-HT, SSRI administration has been associated with decreased sensitivity for rewarding outcomes in some studies[14,15,70]. Reduced reward sensitivity has been attributed to unwanted SSRI treatment effects, notably emotional blunting and reduced efficacy in targeting anhedonia[71]. Importantly, in preclinical work, SSRI administration results in indirect modulation of dopaminergic signalling pathways involved in reward processing[28–31]. In similar work, however, the SSRA used here (low dose fenfluramine, racemic mixture) retains selectivity for 5-HT[45,55,72,73] and is inactive at dopaminergic synapses[56,74], while having a binding affinity for 5-HT transporters which is <0.5% of that typically seen in SSRIs such as citalopram[75] (see the Supplementary Discussion for further details on the past uses

**Table 2 | Mixed-effects linear modelling of longitudinal (daily) Visual Analogue Scale ratings and Side Effects Profile data**

|  | Model estimate [a] | 95% CI | t-value | p |
|---|---|---|---|---|
| Visual Analogue Scale |  |  |  |  |
| Positive items | −17.63 | −67.80, 32.55 | −0.69 | 0.494 |
| Negative items | 28.20 | −8.32, 64.32 | 1.35 | 0.178 |
| Side Effects Profile |  |  |  |  |
| Appetite, decreased | −0.07 | −0.35, 0.21 | −0.50 | 0.622 |
| Appetite, increased | 0.11 | −0.02, 0.24 | 1.74 | 0.089 |
| Drowsiness/Fatigue | −0.02 | −0.23, 0.26 | −0.13 | 0.901 |
| Insomnia | 0.01 | −0.06, 0.23 | 1.18 | 0.242 |
| Sexual side effects | 0.01 | −0.12, 0.14 | 0.14 | 0.888 |
| Sweating | 0.04 | −0.01, 0.09 | 1.48 | 0.147 |
| Tremors | 0.01 | −0.03, 0.06 | 0.57 | 0.571 |
| Agitation | 0.01 | −0.11, 0.11 | 0.05 | 0.959 |
| Anxiety | 0.07 | −0.04, 0.18 | 1.27 | 0.211 |
| Diarrhoea | −0.11 | −0.26, 0.04 | −1.41 | 0.166 |
| Dry Mouth | −0.02 | −0.24, 0.21 | −0.15 | 0.879 |
| Indigestion | 0.05 | −0.01, 0.12 | 1.63 | 0.110 |
| Nausea | −0.05 | −0.20, 0.10 | −0.64 | 0.528 |
| Upset stomach | −0.05 | −0.21, 0.12 | −0.54 | 0.595 |

[a]Main effect of group (active vs placebo) via time-adjusted mixed-effects linear models with restricted maximum likelihood estimation.

of the experimental probe). Thus, these results highlight potentially specific effects of serotonin on loss processing, whereas contradictory effects of SSRIs previously reported may relate to effects beyond the serotonin system. It would be worthwhile to investigate if the effect of the SSRA on aversive processing could prove advantageous for the treatment of depression without exacerbating features of anhedonia.

In the preclinical literature, pharmacological (fenfluramine) and optogenetic stimulation of serotonergic neurons in the dorsal raphe nucleus (DRN) results in no change in reward learning in animal models; however, stimulation of non-serotonergic DRN neurons via amphetamine and optogenetics results in increases in reward choice preference[76]. Moreover, increased firing of amygdala 5-HT projecting neurons is observed during aversive but not reward prediction errors, an effect which appears to be modulated by a functionally discrete DRN to basal amygdala 5-HT pathway[77,78]. Further optogenetic labelling work suggests that increased DRN 5-HT firing promotes aspects of reward processing[79–81], potentially facilitated by co-release of glutamate and subsequent activation of mesoaccumbens dopamine neurons[82]. Given the methodological disparity between this literature and the present work, drawing direct parallels is challenging, particularly as associations between neuronal firing patterns and synaptic serotonin are region-specific[83–87].

Interpreting the ecological meaning of performance during reinforcement learning is challenging. A reduction in sensitivity to loss outcomes may be adaptive or detrimental depending on real-world context. Despite reduced optimal choices for loss trials in the SSRA group, there was no difference in total money earned across groups (see Supplementary Note 3). Moreover, reduced loss learning cannot be attributed to undesired effects such as impaired cognition; indeed, concurrent improvements in learning and memory tasks were observed in the SSRA group.

Increasing synaptic 5-HT (via fenfluramine) enhanced behavioural inhibition, an effect driven by more cautious decision-making. Impairment of 5-HT function decreasing behavioural inhibition is well-observed in animals, and to a lesser extent in humans[9,17]. However, the opposite approach of increasing synaptic 5-HT with SSRIs yields a comparably less clear picture cross-species. In humans, SSRI challenge results in improvement or no change in action cancellation ability (stop signal)[16,88], while action restraint ability (go/no-go) remains unchanged or impaired[17–19,89]. Frontal functional activity increases during action restraint following SSRI challenge, however this is not linked to a corresponding change in ability[18,89]. Likewise, SSRIs yield no clear effect on behavioural inhibition in animals[17,90]. The seemingly irreconcilable effects of SSRIs on behavioural inhibition may be attributed to the vulnerability of the agent to experimental noise; notably, its acute-to-chronic mechanistic shift and off-target dopaminergic effects. Nevertheless, the present study demonstrates objective improvements in action restraint by increasing synaptic 5-HT. Given disorders of behavioural control and impulsivity (e.g., ADHD) are associated with 5-HT dysregulation[90], exploring potential clinical applications of SSRAs within these populations may prove beneficial.

During behavioural inhibition, increased synaptic 5-HT resulted in a bias for impulse control during aversive interference, alongside a corresponding drop in choice impulsivity. This indicates an optimisation in task strategy during aversive interference, which is congruent with increased 5-HT reducing sensitivity to aversive outcomes. These findings align with the longstanding conceptualisation of 5-HT as an inhibitor which becomes active in aversive contexts[91,92]. Indeed, in individuals with depression and tryptophan-depleted healthy adults, choice impulsivity increases for explicit negative emotional targets in a go/no-go paradigm[93–96]. However, the effects of increased 5-HT on behavioural inhibition reported here were not experimentally confined to aversive contexts; notably, we observed a decrease in choice impulsivity during a control condition without affective interference. Potentially then, 5-HT performs an active role of limiting impulsive action across emotional and neutral contexts, but this is amplified in aversive contexts.

Increasing synaptic 5-HT also enhanced retrieval and speed of processing during memory tasks involving verbal information. Observable changes in verbal but not visuospatial learning is reliably observed following TRP depletion[12]. The effects of tryptophan depletion on complex verbal working memory are less clear, owing to insufficient studies in this area. SSRI challenge, however, leads to highly variable effects on memory function; while improvements have been observed, typically null findings are reported[21,22,97]. Unlike fenfluramine, the threshold of synaptic 5-HT required for observable change may not be achieved during the brief SSRI regimen of most studies ($\leq 7$ days), where the problem of autoreceptor supersensitivity persists[39,98]. Importantly, 5-HT receptor subtypes strongly associated with memory functioning (i.e., 5-HT$_{3,4,6}$ receptors) have significantly lower binding affinities for endogenous 5-HT relative to other 5-HT receptors (e.g., 5-HT$_{1A,B,D,E,F}$; 5-HT2$_{A-C}$)[99–102]. Thus, crossing a putative 5-HT concentration threshold may be required to observe change in memory function, potentially explaining our findings.

As with most psychoactive substances, including SSRIs[103–107], the SSRA fenfluramine may produce ancillary off-target pharmacological effects alongside its primary mechanism. While retaining neurotransmitter selectivity for the serotonergic system at low doses, fenfluramine has modest affinity for the 5-HTR subtypes and sigma-1 ($\sigma_1$) receptors. In the case of 5-HTR, these effects appear to be specific to 5-HT$_{2A/B/C}$R[108], while there is inconclusive evidence of the involvement of other receptors, such as 5-HT$_4$[109]; however, the binding affinity of fenfluramine for 5-HT$_{2A/B/C}$R is at most <1% of that of competitive endogenous 5-HT[100,108,110]. Indeed, given the high concentration of 5-HT following exocytic release and the finite availability of 5-HT receptors[38,55,111], the resulting ancillary effects of 5-HTR agonism/antagonism may be negligible. Moreover, fenfluramine appears to be both a positive allosteric modulator and antagonist of $\sigma_1$R[112,113]; while endogenous neurosteroid sigma-1 agonists may be potentiated through positive allosteric modulation, leading to improvements in cognitive ability[113–115], it is unclear to what extent this effect may be

offset by the sigma-1 antagonist properties of fenfluramine in the healthy brain[112,115–117].

Pharmacodynamic data on norfenfluramine, the neuroactive metabolite of fenfluramine, is limited; in one study, *d*-norfenfluramine administration in mice resulted in small increases (relative to 5-HT) of synaptic noradrenaline[118]. In two further studies, however, noradrenaline levels were unaltered by *d*-norfenfluramine and *dl*-norfenfluramine administration[119,120]. Moreover, *dl*-fenfluramine administration in humans produces plasma concentrations of *dl*-fenfluramine and *dl*-norfenfluramine at a 1:3 ratio, respectively, with only a fraction of that being *d*-norfenfluramine[118,121]. Nevertheless, contrasting the neurobehavioural profile of SSRA fenfluramine with *S*-enantiomers (selective to 5-HT) of SRAs such as 4-methyl-N-methylcathinone[39], once clinically available, could provide further insight. Beyond this, while there is human evidence of in vivo serotonergic modulation by *d*-fenfluramine with [^18F]altanserin PET[57], further human PET/SPECT investigations of low dose fenfluramine may offer additional insights into its neurochemical profile. Finally, when interpreting the present findings in the context of past studies of TRP, worth noting is the disputed contribution of non-serotonergic modulation via the kynurenine pathway from TRP[107,122].

Methodological differences across studies complicate the interpretation of findings, past and present, within a broader literature context. While earlier we mentioned key differences in response inhibition paradigms (i.e., Stop Signal vs Go/No-Go tasks), paradigms which capture reinforcement learning in humans also splinter in a manner which hinders direct comparison. Notably, reinforcement learning paradigms with or without components such as reversal or model-free/model-based learning involve different computational models and neural pathways[64,92,123–127]; consequentially, model-free learning implemented in the present work is challenging to compare with TRP/SSRI work involving model-based and reversal learning[128–130]. For example, previous work suggests 5-HT depletion (via pharmacological lesioning) modulates reinforcement sensitivity to misleading punishments and rewards during reversal learning[131]. It is important, therefore, to interpret the present findings within the context of its methodological fit to past literature.

In summary, we demonstrate direct effects of increased synaptic serotonin on human behaviour, underlining its role in guiding decision-making within aversive and neutral contexts. In aversive contexts, increased synaptic serotonin appears to reduce sensitivity for loss outcomes, and promotes a bias toward impulse control during behavioural inhibition. In neutral contexts, increased synaptic serotonin appears to enhance behavioural inhibition by promoting cautious decisions, as well as enhancing memory recall for verbal information.

Not only do the present findings offer implications for long-standing theories of central serotonin, but they also demonstrate the promise of the SSRA as an experimental probe, furthering the scope of fundamental work which aims to characterise the involvement of serotonin in human behaviour, and its contribution to psychiatric aetiology in clinical samples.

Given the prominence of impaired cognition and aversive/negative emotional biases as transdiagnostic targets within psychiatry (e.g., unipolar and bipolar depression; schizophrenia)[21,71,132–135], investigating the therapeutic potential of the SSRA in clinical populations may be worthwhile. Such investigations may allow greater targeting of specific neurocognitive mechanisms across disorders in the absence of widespread, and often unwanted, effects including emotional blunting.

## Methods
### Participants and design
Fifty-six participants (28:28, SSRA:placebo; mean age = 20.2) were randomised to take part in the study. Recruitment occurred between June 2021 and June 2022. Potential participants were screened to exclude those who had recently used recreational drugs (3-month wash-out,

except MDMA which had a wash out period of $\geq 1$ year), who were pregnant, trying to become pregnant, or who were currently breastfeeding. All participants had a BMI between 18–30 and were fluent speakers of English. For full exclusion and inclusion criteria, please see Supplementary Methods. For full details of the recruitment process, see the study CONSORT flow diagram (Fig. 7). The final sample consisted of 53 young, non-clinical participants (mean age = 20.15; 32 female) which were allocated to administration of serotonin releasing agent fenfluramine ($n = 26$) or placebo ($n = 27$) for subchronic administration ($8 \pm 1$ days). All participants in the final sample attended sessions before treatment and at follow-up. Participants were reimbursed 175 GBP upon completion of the study. Participants were requested to self-report on gender, and the potential covariance of gender with treatment effects was investigated post-hoc. The detailed results of these analyses are provided in the Supplementary Tables 5–7.

Eligible participants were randomised to administration of SSRA fenfluramine hydrochloride (15 mg b.i.d.; racemic mixture) or placebo for subchronic administration ($8 \pm 1$ days), in a double-blind design. Both fenfluramine and the placebo were administered orally in a flavoured aqueous solution, with the placebo lacking an active pharmaceutical ingredient. Randomisation was performed by the Clinical Pharmacy Support Unit, Oxford Health NHS Foundation Trust (Oxfordshire, United Kingdom) using a stratified block randomisation algorithm, with stratification for gender and task stimulus version (for further details on task stimulus versions, see Supplementary Methods).

The study was approved by the University of Oxford Central University Research Ethics Committee (MSD-IDREC reference R69642/RE004) and pre-registered on the National Institute of Health Clinical Trials Database (NCT05026398). All primary outcomes within the pre-registered are reported within the current paper. Prior to study participation, participants provided informed consent. All study visits were conducted at the Department of Psychiatry, University of Oxford.

### Procedure
Participants undertook two screening visits to assess study eligibility. In the first session, medical history and current medication use was assessed and the Structured Clinical Interview for DSM-V was conducted to screen for current or past psychiatric illness. In the second session, cardiovascular health (blood pressure; electrocardiography), renal and liver health (liver function, urea, and electrolyte blood tests) were assessed, and drug and pregnancy urine tests were performed. Eligible participants attended two study visits, baseline and post-intervention occurring 7, 8 or 9 days later. This study period was scheduled to avoid the premenstrual week for female participants. At baseline, participants completed a battery of cognitive and emotional computer tasks and questionnaires. Participants were then given their first dose of the SSRA or placebo, and monitored for three hours during which regular blood pressure and observational checks were made. Measurements of early changes in salivary cortisol concentration were taken to investigate potential modulation of HPA axis function that has been previously observed in studies of acute high dose fenfluramine[136]. Saliva samples were collected immediately before initial dose, one hour post-dose, and three hours post-dose. These samples were immunoassayed for cortisol levels over linear calibration curves (for further details, see Supplementary Methods). After the initial dose visit, participants were asked to independently take the fenfluramine or placebo daily, in addition to completing daily questionnaires (see Questionnaires measures section). At the post-intervention visit, participants completed the same task and questionnaire battery as at baseline and were then requested to estimate their allocation prior to debriefing.

### Questionnaire measures
At each study visit, participants completed self-report questionnaires measuring affect, mood, anxiety, subjective cognitive functioning, and

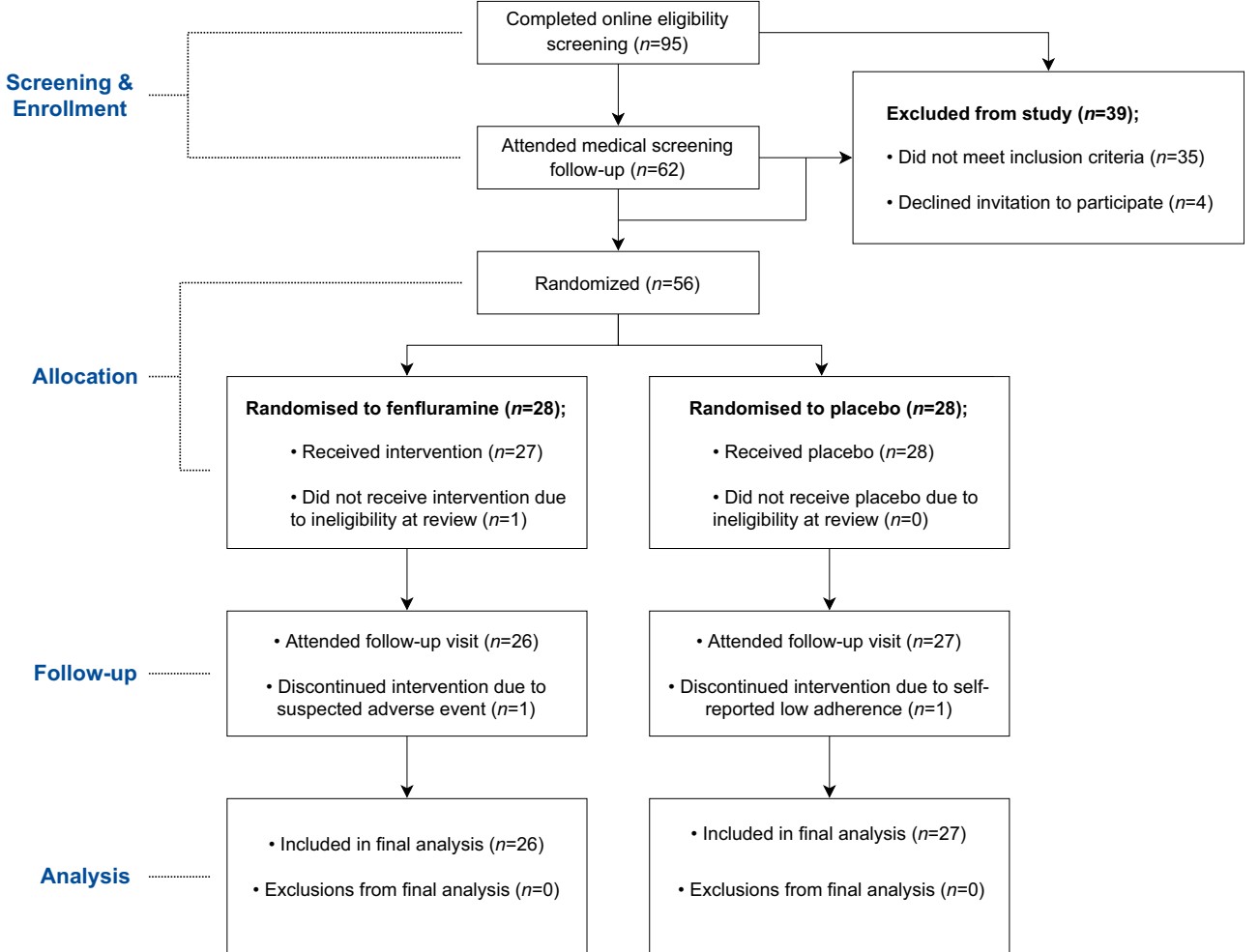

**Fig. 7 | CONSORT Diagram of participant flow throughout the study.** The flowchart shows the study recruitment process, which consisted of two screening visits (online and in-person) and two study visits (an initial dose visit and follow-up visit occurring 8 ± 1 days later). Eligible participants were invited to participate in the study and, upon acceptance, were randomised to receive either fenfluramine or placebo.

side-effects; the Spielberger State-Trait Anxiety Inventory [STAI-T], Beck Depression Inventory II [BDI], Positive and Negative Affect Schedule [PANAS], Visual Analogue Scale [VAS], Perceived Deficit Questionnaire – Depression [PDQ-D], and side effects profile questionnaire. Participants completed the VAS and side effects questionnaire once per day from the baseline visit until the follow-up visit.

**Cognitive and Emotional Task Battery**

Participants undertook an extensive cognitive and emotional task battery at both the initial dose visit (baseline) and follow-up visit. Participants undertook the following tasks in order: 1) Auditory Verbal Learning Task (AVLT) (Fig. 6D) – a measure of declarative memory encoding and retrieval where accuracy of recall was the measured outcome. Behavioural data for the task consisted of $N = 51$ individuals (mean age = 20.15; 31 female). 2) Affective Interference Go/No-Go Task (Fig. 3A) – a measure of behavioural inhibition under affective interference (positive [happy faces], aversive/negative [fearful faces], and neutral distractors) where accuracy of inhibited response to no-go trials (response inhibition), accuracy, and response time for go trials (an index of impulsivity[137]) were the non-model outcome measures. The block design of the task allows for analysis of set-shifting effects (executive shifting for task condition rule changes) on accuracy and response time. Participant task data was fit to a computational drift diffusion model (see Supplementary Methods for further details) which provided the following model parameters: boundary separation, initial choice bias, non-decision time, drift rate, and drift criterion. Behavioural and computational data for the task consisted of $N = 50$ individuals (mean age = 20.22; 32 female). 3) Verbal n-back task (Fig. 6A) – a measure of complex verbal working memory where accuracy and response time to target letters (i.e., matching a letter which appeared n-back [0, 1, 2, or 3] trials ago) were the outcome measures. Behavioural data for the n-back consisted of $N = 50$ individuals (mean age = 20.1; 31 female). 4) Probabilistic instrumental learning task ([Fig. 2A]; adapted from[63]) – a measure of reward and loss sensitivity during instrumental learning, which produced non-model outcome measures which were fit to computational reinforcement learning models. Non-model outcomes were optimal choice outcome (i.e., selecting the stimulus with a higher probability of a favourable outcome under each task condition: wins during win trials and no change during loss trials) and response time. The main computational model parameters were outcome sensitivity and learning rate, where learning rate was estimated separately for both win and loss trials (see Supplementary Methods for further details). Behavioural and computational data for the task consisted of $N = 53$ individuals (mean age = 20.15; 32 female). 5) Oxford Memory Task – a measure of visuospatial working memory which included localisation speed and stimulus selection accuracy outcomes. Behavioural data for the task consisted

of $N = 51$ individuals (mean age = 20.16; 31 female). 6) Contextual cueing task – a measure of implicit learning and visual search ability where the outcome measure was accuracy and response times under novel/implicit cueing conditions. Behavioural data for the task consisted of $N = 53$ individuals (mean age = 20.15; 32 female). Full details of tasks included in this battery are included in the Supplementary Methods.

## Statistical analysis

Data pre-processing and statistical analyses were carried out using R Software (version 4.3.1), and computational modelling was undertaken using MATLAB (R2022a), Docker (version 4.22.0), and Python (version 3.8.8); required software dependencies and packages are listed within the Supplementary Methods. Homogeneity in demographic variables across allocation groups was assessed using chi-squared independence tests (categorical, binary variables) and two-tailed independent t-tests (continuous, discrete variables). The effect of the SSRA on outcomes across the task battery and questionnaire ratings was analysed using between-groups (SSRA vs. placebo) mixed effects ANCOVA models (two-tailed) on post-intervention data, with baseline visit performance serving as a regressor and participant as a random effect for nested multilevel data structures (e.g., multiple task conditions). The approach of using baseline score as a regressor in this manner was selected as it allows isolation of the group effect from potential sources of bias (e.g., learning and practice effects)[138,139]. In comparison with other baseline-adjustment techniques, such as change score between before and after intervention, baseline-adjusted ANCOVA yields greater statistical efficiency and precision (irrespective of baseline balance/imbalance)[140,141]. Planned comparisons were carried out on outcome measures collected at follow-up using two-tailed estimated marginal means tests, where estimates are reported alongside standard mean error. Family-wise error for estimated marginal means tests was adjusted using the Bonferroni-Holm procedure. Models were assessed through histogram examination of standardised residuals to determine normality of distribution (non-normal computational values were log-transformed), and homogeneity of regression slopes and covariate independence were tested by modelling covariate interactions across factors (covariate × group × task condition). Two-tailed effect sizes metrics are reported for significant ANCOVA (partial eta squared, $\eta_p^2$) and EMM (Cohen's $d$, $d$) models alongside corresponding 95% confidence intervals (for effect size calculations, see Supplementary Methods). Post-hoc correlational analyses on the Affective Interference Go/No-Go task were performed using two-tailed Pearson's product-moment correlation analysis. Baseline-adjusted ANCOVA models were generated post-hoc to investigate the potential covariance of gender with group effects across study outcomes, which are reported in Supplementary Tables 5–7. Group allocation guesses were analysed using chi-squared independence tests. In addition to ANCOVA analysis of questionnaire data at follow-up, daily questionnaire data (VAS and side effects profile) was joined longitudinally with initial dose and follow-up visit data and analysed using linear mixed effects models with restricted maximum likelihood estimation where time served as a regressor. Salivary cortisol was analysed across three timepoints (before dose, 1- and 3-hr post-dose) using linear mixed effects modelling with time as a regressor. Baseline task battery, cortisol and self-report questionnaire data are included in Supplementary Figs. 5, 7–9, 11–13, 16, 18, 20–23, 25. All inferential analyses were carried out at the 0.05 alpha level. A priori sample size calculation (Power [1 - β error probability]: 80%) determined that a sample of $N = 52$ (26 per group) was required to undertake two-tailed between-groups analysis.

## Reporting summary

Further information on research design is available in the Nature Portfolio Reporting Summary linked to this article.

## Data availability

Raw and modelled datasets generated for this study have been deposited on Zenodo[142] and Github: https://github.com/mjcolwell/SSRA_human_behaviour_data_and_code.

## Code availability

The code used to undertake data preprocessing, computational modelling, and inferential analyses in this study has been deposited on Zenodo[142] and GitHub: https://github.com/mjcolwell/SSRA_human_behaviour_data_and_code.

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

## Acknowledgements

This study was funded by a grant from Zogenix International Ltd. (awarded to C.H.), prior to merge with UCB Pharma, and supported by the NIHR Oxford Health Biomedical Research Centre and the NIHR Oxford Cognitive Health Clinical Research Facility. The views expressed are those of the authors and not necessarily those of the NHS, the NIHR or the Department of Health. We thank Dr Sandra Tamm, Dr Angharad de Cates, Dr Sara Costi, and Dr Alexander Smith for their assistance with medical screening and diagnostics procedures. We thank Prof Valerie Voon for suggestions for data analysis. We thank Dr Margarita Chibalina for their assistance with biological sample handling and processing. We thank Dr Jan Willem de Gee for publishing openly available computational modelling scripts. We thank Tara Pusinelli for assistance with data collection and entry. We thank Sorcha Hamilton for help with verifying code reproducibility.

## Author contributions

M.C., S.M., C.H. and P.C. designed the study. M.C., H.T., H.I.C. and C.E.W. undertook data collection. M.C., H.T., H.I.C. and M.B. produced preprocessing task scripts. M.C. and C.E.W. undertook biological specimen processing. F.S. and M.B. provided computational modelling support. M.C. undertook all computational modelling and inferential analyses. H.T. and F.S. contributed equally to this work as second authors. M.C., S.M., C.H., P.C., and M.B. undertook data interpretation. M.C. drafted the article and produced illustrations. All authors contributed to revisions and approval of the final draft.

## Competing interests

C.H. has received consultancy fees from P1vital Ltd., Janssen Pharmaceuticals, UCB Pharma, Sage Therapeutics, Pfizer, Zogenix, Compass Pathways, and Lundbeck, and is a director of the company TnC Psychiatry and Neuroscience. S.M. has received consultancy fees from Zogenix, Sumitomo Dainippon Pharma, P1vital Ltd., UCB Pharma and Janssen Pharmaceuticals. C.H. and S.M. hold grant income from Zogenix, UCB Pharma, Syndesi and Janssen Pharmaceuticals. C.H. and P.C. hold grant income from a collaborative research project with Pfizer. M.B. has received travel expenses from Lundbeck for attending conferences and has acted as a consultant for J&J, Novartis, Boehringer and CHDR. He previously owned shares in P1vital Ltd. The other authors report no conflicts of interest.
