## [Peer Review File · Nature Communications]

REVIEWER COMMENTS

Reviewer #1 (Remarks to the Author):

This article is a psychopharmacological investigation of the effects of subchronic fenfluramine, a 5-HT releasing agent. Fenfluramine was previously employed as a treatment for obesity and also autism (with no real efficacy). However, its use was discontinued through cardiovascular side-effects. Recently however it has been employed with success in the treatment of Dravet's syndrome and the present authors have capitalised on this as a relatively 'new' means of probing the 5-HT system in healthy humans, as there has been relatively little analysis of its effects on cognition hitherto. Such an analysis may thus help to resolve questions about the role of 5-HT in behavioral and cognitive function in view of some mixed conclusions following effects of serotonin reuptake inhibitors. The authors have studied the effects of a single dose of fenfluramine on reinforcement learning, 'behavioral inhibition', working memory and verbal long term memory. The study uses a parallel between-group, placebo-controlled design and is (presumably) double blind. Both male and female volunteers were recruited. They employ a sophisticated behavioral test battery and data analysis procedures. The most clear-cut findings were selective effects on punishment/aversive processing compared with reward, enhanced inhibition and improved verbal long term memory. The authors use these findings to claim resolution of basic issues about the role of 5-HT in these processes and to argue for a possible alternative clinical strategy for enhancing 5-HT function to the SSRIs.

Critique. The authors are assuming that their regimen of fenfluramine administration is producing selective boost to 5-HT release. Preclinical studies are consistent with this; however, human studies with PET are less in evidence, especially on sub-chronic administration. It is therefore a limitation that the authors are unable to show effects of the drug consistent with changes in human brain. In an extended Supplementary discussion the authors also document other known effects of fenfluramine, such as its action at sigma receptors. These limitations on the conclusions are not discussed in the main body of the ms. In order to bolster their conclusions regarding fenfluramine, the authors rely on comparisons with effects of dietary tryptophan depletion. Although some of these are indeed opposite in sign and therefore congruent with the present findings, the authors are often quite selective in their choice of which comparisons are made to favour their own findings. The same criticism applies to the use of the preclinical literature which is occasionally mis-cited. Hence, although this new study provides a useful new approach to the problem of 5-HT function, with some new findings, the conclusions are incremental rather than definitive.

Specific points

1. Overall, the rationale of this new approach, reasonably enough, is to help clarify some of the apparent discrepancies found in previous studies, especially of effects of SSRIs. However,

some of these 'discrepancies' probably result from such factors as duration of chronic treatment or differences in test paradigms for some cognitive constructs- and perhaps highlight the need for parametric exploration of these issues. It should be noted however that the present study also suffers from some of these same considerations, which should be acknowledged.

2. The neural effects of sub-chronic fenfluramine are documented for rodents and not humans. This should perhaps be made clearer in the main text.

3. Methods. The rationale for the between-groups rather than cross-over study is not explained very clearly. The function of the baseline testing is also not made clear. Are there findings based on change scores? Was the study double blind?

3. Behavioral tests. Is it completely accurate to call the verbal learning test one of "episodic memory" (in the sense of what, where and when?).

4. Test of behavioral inhibition. It should perhaps be acknowledged that 'tests of behavioral inhibition' often in fact probe different processes. For the stop-signal task, another standard test of this construct, acute dietary tryptophan depletion does not affect performance in healthy volunteers (Clark et al Psychopharmacology 2005)- which is obviously not consistent with the conclusion of a "generalised" role for 5-HT in 'behavioral inhibition'. On the other hand, in a Go-NoGo test tryptophan depletion only induced disinhibition in the punishment condition (Crockett et al J. Neuroscience 2009). The latter is presumably consistent with the authors' hypothesis of selective valence effects of fenfluramine, although it is not cited. However, it should also be acknowledged that there are the limitations of using tryptophan depletion as a robust criterion of 5-HT deficiency, especially when the findings are

anti-thetical to the observed findings here.

5. Test of instrumental learning. It was not quite clear to me how the best fitting model was derived. Which other competing alternatives were considered and how was the winning model chosen? Details should be provided.

6. Not sure that Fig 4 and its accompanying elegant drift diffusion analysis in fact adds much to the overall understanding. Perhaps the legend needs to be more explicit.

7. Was sex/gender considered as a factor for analysis? Perhaps merits some comment.

8. The relevance of the cortisol test and findings was not made clear.

9. Conclusions. (i) Asymmetric effects of valence following tryptophan depletion have also been shown for model based versus model free processing (Worbe et al Mol Psychiat 2017).

(ii) The claim that the preclinical, optogenetic literature supports the selective effects on aversive, as distinct from reward, processing is controversial to say the least. The paper by Rygula et al on my reading in fact seems to conclude no selective effects on aversive learning rate. I think these claims may need to be watered down....

(iii) The apparent conclusion that verbal memory retrieval was selectively affected over non-verbal memory is also apparently inconsistent with previously published effects of tryptophan depletion

(iv) Despite their literature review in the Supplementary Material, the authors fail to cite the important article by Luciana et al (1998 Cerebral Cortex) on effects of fenfluramine on working memory in humans. An acute dose of 60mg impaired spatial working memory which perhaps should be commented upon.

10. Typo I think that serotonin is missing from the initial sentence of the Discussion (p15)

SSRA- not sure this is yet fully justified as an abbreviation rather than referring simply to fenfluramine. This not yet an acknowledged category of psychoactive drugs.

Title of the ms. Seems a little pretentious, yet uninformative? "Selective effects of serotonin release on cognitive and aversive processing in humans" may be a more accurate and encompassing description of findings.

Reviewer #2 (Remarks to the Author):

This is a very strong paper. It is concisely and clearly written, and well embedded within existing literature on serotonin's functional contributions. It reports the cognitive effects of a clinically highly innovative pharmacological probe (the selective serotonin releasing agent fenfluramine) which is mechanistically more potent and more precise than SSRIs, not suffering from the off-target dopaminergic effects or the acute-to-chronic mechanistic shift known to occur for SSRIs. As a result, it enables the authors to test more cleanly various predictions from longstanding theories of brain serotonin function. The paper neatly summarizes the results of a study in which 2 groups of healthy volunteers (27 vs 28) were given a low oral dose of fenfluramine for 7-9 days in a row, in a double-blind design. The results are fascinating, substantiating previous hypotheses about serotonin's role in behavioral inhibition, aversive slowing and memory enhancement. The paper represents a major contribution to the literature and will stir lots of new research, both basic science and clinically applied.

I have the following suggestions:

- To clarify the key features of the study design (subchronic, 7-9 day oral administration, between-subjects, numbers of subjects per group!, pre- vs post-drug testing) at the beginning of the results section. Only after reading the methods section at the end of the paper did it become clear that the study was a between-group, pre/post-test design, involving intake of fenfluramine for multiple consecutive days.

- To more clearly justify the approach of using baseline performance as a regressor rather than adopting more generally employed/accepted repeated-measures ANOVAs or linear models. We were not sure we understand why the chosen approach would yield greater statistical efficiency

and what the authors meant with the statement that this avoids conditional bias from baseline imbalance. This seems pertinent because the authors did choose to use the more classic linear modeling approach when analyzing cortisol and VAS and side effects.

- Regardless of the above, I strongly recommend full reporting and visual presentation of all the data, including the baseline data.

- To also plot, for full reporting purposes, drug effects on the win trials (not just the significant loss parameters) in the main text

- To report the effect of drug on response inhibition (% nogo) as a function of valence (or task interference: happy, fearful or control distractors) (analogous to the RT data analysis for this task). In the current version of the MS, fenfluramine is reported to increase response inhibition across all conditions. Can the stats be presented and data be plotted as a function of valence (or: 'task interference')? This will allow the reader to evaluate the degree to which the effect on response inhibition is indeed generalized or valence specific.

- To clarify, justify, and possibly revise the interpretation of the response time effects in the go/nogo task in terms of improvements in impulse control (or enhanced action restraint). What aspect of this task or of the performance pattern makes the authors interpret the slowing of RTs as evidence for greater impulse control? I am not convinced the GO RTs represent impulsive choices in the first place. It seems equally possible that fenfluramine brought to the surface, rather than restrained impulsive behavior. After all, slowing on aversive trials has previously been conceptualized as an aversive Pavlovian (hardwired) inhibition 'impulse'. If the authors agree, then this would require some revision throughout the general discussion of the paper, which is referring to improved impulse control in various places. If they don't, then it would help to make more explicit the evidence for faster RTs on this task reflecting choice impulsivity.

- Might the drug-related attenuation of loss sensitivity during learning reflect an indirect consequence of this increase in aversive Pavlovian inhibition? In this context, might it be informative to compute correlations between the effect on aversive GO RTs and the effect on loss learning.

- To move a variety of key methods section from the supplement to the main manuscript. For example, the subjective VAS and side effect data are key, given that this is one of the first reports of this compound. Furthermore, the reader would benefit from access to some more modeling details to evaluate the computational modeling procedures. What was the model (space)? Was any model comparison performed? Were simulations performed? Isn't there redundancy when modeling both

outcome sensitivity and decision temperature? Why include the outcome sensitivity parameter? Is this the same choice stochasticity parameter that was previously found to be modulated by serotonin manipulations (e.g. acute tryptophan depletion by Ben Seymour et al)?

- To clarify whether a power calculation was performed to justify the sample sizes.

Minor details:

I would suggest breaking down the group x task interaction for GO response times in a way that makes the nature of the interaction more obvious. Was there perhaps a simple main effect of valence (aversive vs control) for drug, but not placebo? And what about positive vs control?

Page 15, line 273: is the word '5-HT' missing after 'synaptic'?

Page 16, line 308: is the word '-projecting' missing after '5-HT'?

Reviewer #3 (Remarks to the Author):

In their manuscript „Direct serotonin release in humans shapes decision computations within aversive environments“, Colwell et al. use behavioural testing, computational modelling and pharmacological neuromodulation to examine the impact of repeated oral administration of a serotonergic agent (the SSRA fenfluramine) on cognitive measures including learning, behavioural inhibition and memory. The authors found that serotonin reduces sensitivity to aversive outcomes during learning, enhances behavioural inhibition, and modulates memory for verbal information. Overall, the study is well-executed, the manuscript is well-written, and the results are important. I believe, the findings advance the canon of knowledge in the field. However, I think the following points could be clarified and improved in the manuscript.

1) As far as I understand, the authors tested subjects twice, once pre-treatment, and once after/during treatment. They use baseline score as regressor in ANOCVA. This is critical. It seems to me the authors have thought about this carefully. However, I feel, it'd be important to back this approach up with some data. I think, the least the authors should do is show the baseline data. The authors state by citing references that this is in their opinion the best way to analyse the data. However, they refer to clinical trials. This is not trivial, given cognitive neuroscience trials using

cognitive tasks (as in the current study), feature characteristics that clinical trials may lack, e.g., learning effects when testing subjects repeatedly. Thus, referring to clinical trials may not suffice when making this important analytical decision. Relatedly, what is the advantage of having a baseline session in the first place, or in other words, what was the rationale for the study design? Because familiarising the participants with the task can also be done in other ways, e.g., training before the actual task session. Thus, it'd be important to know more about the decision process re. the study design and analytical approach? Further, this analytical approach should be more clearly mentioned in the results section (similar to the supplement, where the authors state "baseline-adjusted ANCOVA").

2) The authors mention that baseline performance served as a regressor and participant as a random effect "where appropriate" (cf. Supplementary Material). Here, I think, it is important to clarify what "where appropriate" means?

3) FWE correction is mentioned in the methods section. However, where was this applied in the actual analysis? Given the many analyses, and the lack of clearly defined hypotheses for many cognitive tasks reported, this seems important.

4) In my view, there is missing detail on the methods of the learning task. For instance: What are the number of trials, number of stimulus pairs, number of blocks in the task?

5) Further, if I am not mistaken, the authors mention they only analysed the last 40 trials of each block (cf. Supplementary Material). I assume, performance then is stable (which makes sense when analysing choice behaviour), however, I also assume a major bulk of learning is taking place before that? Given the authors use a RL learning model, I don't fully understand why the authors take this approach, and I wonder whether the authors can elaborate on this approach a bit more?

6) The authors (in the Supplementary Material) mention "total gained" as a performance measure in the methods. However, I don't find the results in the results section. Given performance is different between drug groups (worse, as it seems, for the SSRA), it'd be interesting to have some kind of measure of overall performance (which may, or may not be, the authors can clarify here, total points gained)?

7) The authors mention that the SSRA reduced the number of optimal choices during loss trials only (cf. Fig. 2D). As mentioned above (cf. comment #3), I wonder whether the authors can elaborate on this result a bit more, as it seems this relates to worse performance in the task? If so, I believe this is worth mentioning and discussing, also, but not only, because performance on other tasks (cf. below) is improved.

8) Supplementary Material, p3, line 125-126. The description of the outcome coding is not clear to me. Shouldn't it be "0" in loss trials instead of "no change", and "-1" in loss trials? The coding is critical when analysing and interpreting valence (positive/negative outcomes) effects.

9) "The unchosen option was not kept constant but rather updated based on the reciprocal outcome", cf. Supplementary Material p4. Maybe the authors can elaborate on this approach for the reader to understand? If task design is explained more detailed (cf. above), then this should be much clearer.

10) The authors mention (and show, cf. Supplementary Material) a "perfect correlation" between the decision temperature and outcome sensitivity parameters. This makes me wonder: Why did the authors not compare different models with different sets of parameters? It is unclear to me why the authors opt for this model, given it seems that different parameters are capturing the same effects/processes, and are, as it seems, redundant. Such problems can be addressed with, for instance, model comparison, of different models with different sets of model parameters. Given the authors make strong claims based on the "computational" results, this seems critical to me. Given also that the model-free results are convincing and strong, there is no doubt that that the modelling results can reveal important mechanistic insights, however, as is, I find the interpretation of the modelling results rather difficult.

11) Related to the prior comment: what is the difference between the learning rates and outcome sensitivity parameters, and what is the rationale for having both parameters, and on top of that, having them separate for win/loss trials? What is the rationale for having so many parameters, are different parameters needed for win/loss trials? I believe that such questions can be addressed by model comparisons, for instance, having different models that have separate learning rates for win/loss, and shared learning rates, etc. are needed.

12) Related to the prior comments: I believe, the authors need to present simulated data (from the winning models) and model recovery results (for the parameters) as is common practice in cognitive computational research to make sure the models are actually capturing what the authors claim they do.

13) The results for the behavioural inhibition task – if my interpretation is correct – show that inhibitory processes (no-go) are most impacted by the SSRA when aversive distractors are present. It'd be interesting to have some interpretation of this result, given it shows that subjects were more sensitive to aversive stimuli, which seems to contrast with the learning results, where subjects were less sensitive to aversive stimuli?

14) Related to the prior comment: is it fair to say that SSRA subjects performed better in the behavioural inhibition task? And is that different to the learning results (cf. comments #3 and #4). This is by no means a methodological critique, it'd just be interesting to hear about the authors' interpretation of these results.

15) Did I understand correctly that subjects took the drug repeatedly (daily) for multiple days? If so, I think, this is (one of the many) strength of this study, and I think, the authors could more prominently inform the reader about this in the abstract, introduction, and results (as it is, it is "hidden" in the methods only).

16) Similar to my prior comment, I believe, the authors should consider placing the results of the placebo/drug allocation guess in the results section rather than put it in the methods only. I think, this is an important result that should be mentioned in the main manuscript, partly (not only) because it addresses an often-raised criticism of clinical trials in psychiatry on, e.g., antidepressant drugs. However, this is up for the authors to decide, no need to please the reviewer here, but rather a suggestion.

17) In the discussion and conclusion section, the authors mention "more neutral contexts". I think, it would help if the authors clarified this, as it is not clear to me what this is supposed to mean.

18) Given the robust and important findings, I think, it is justifiable for the authors to at least – albeit cautiously – discuss their findings with re. to implications for treatment of psychiatric disorders. It'd be nice to hear some of the author's ideas about how these results relate to their own prior extensive work (e.g., on SSRIs) and discuss clinical implications.

19) Supplementary fig. 2/3 is wrongly referenced in supplementary methods, e.g., p6 line 223/227; the respective figures do not show the referenced results. So there seems to be a mix-up here.

20) p5, line 99: "historically considered historically core functions of serotonin". Consider revising this sentence.

21) Supplementary Material, p5, line 210: Results "bordered" significance. Further, the reader is not informed about the direction of results. I think that showing "bordered" (not corrected for multiple comparisons?), non-significant results sort of renders using alpha levels of no avail.

Responses to Reviewer #1

1.1 Remarks to the Author: *"This article is a psychopharmacological investigation of the effects of subchronic fenfluramine, a 5-HT releasing agent. Fenfluramine was previously employed as a treatment for obesity and also autism (with no real efficacy). However, its use was discontinued through cardiovascular side-effects. Recently however it has been employed with success in the treatment of Dravet's syndrome and the present authors have capitalised on this as a relatively 'new' means of probing the 5-HT system in healthy humans, as there has been relatively little analysis of its effects on cognition hitherto. Such an analysis may thus help to resolve questions about the role of 5-HT in behavioral and cognitive function in view of some mixed conclusions following effects of serotonin reuptake inhibitors. The authors have studied the effects of a single dose of fenfluramine on reinforcement learning, 'behavioral inhibition', working memory and verbal long term memory. The study uses a parallel between-group, placebo-controlled design and is (presumably) double blind. Both male and female volunteers were recruited. They employ a sophisticated behavioral test battery and data analysis procedures. The most clear-cut findings were selective effects on punishment/aversive processing compared with reward, enhanced inhibition and improved verbal long term memory. The authors use these findings to claim resolution of basic issues about the role of 5-HT in these processes and to argue for a possible alternative clinical strategy for enhancing 5-HT function to the SSRIs."*

1.1 Author Response: We would like to thank the reviewer for taking the time to provide such thorough feedback on our manuscript. We have worked to integrate this feedback to the revised manuscript; below we detail changes made in response to your review and provide further clarification on points you have made.

1.2 Remarks to the Author: *"Critique. The authors are assuming that their regimen of fenfluramine administration is producing selective boost to 5-HT release. Preclinical studies are consistent with this; however, human studies with PET are less in evidence, especially on sub-chronic administration. It is therefore a limitation that the authors are unable to show effects of the drug consistent with changes in human brain."*

1.2 Author Response: Thank you for this comment. It is certainly correct that preclinical studies outnumber work in humans on the effect of fenfluramine on synaptic 5-HT levels. Multiple human PET studies show the influence of fenfluramine on correlates of neural activity, however much of this work is limited to radioligands which track systems non-specific to serotonin binding (e.g., [¹⁵O]H₂O and [¹⁸F]fluorodeoxyglucose)^{1,2}. However, one notable study utilised [¹⁸F]altanserin, a radioligand selective to 5-HT_{2A}, which is capable of estimating changes in serotonin release across the brain^{3,4}; within this study, the authors found that acute and subchronic [14 days] administration of *d*-fenfluramine

dose-dependently decreased volumes of the radioligand ³. These findings are suggestive of increased synaptic serotonin release in the human brain related to fenfluramine. We, however, acknowledge that further PET studies on dl-fenfluramine, using the regimen in our study, would indeed be beneficial. We have added this point as a limitation within our discussion.

Regarding its selectivity for the 5-HT system at lower doses, this is only feasibly derived from preclinical work at the current time. As a general point, it is highly difficult to determine the selectivity of a psychoactive substance to a particular neurotransmission system in humans. Namely, there are inherent issues of availability and selectivity for many neuromodulator PET ligands ^{5,6}, a problem which does not apply to the *in vivo* microdialysis work we have cited.

1.3 Remarks to the Author: *"In an extended Supplementary discussion the authors also document other known effects of fenfluramine, such as its action at sigma receptors. These limitations on the conclusions are not discussed in the main body of the ms."*

1.3 Author Response: In response to this feedback, we have now moved this section of the Supplementary Discussion to the Discussion within the main manuscript (see lines 402 – 423).

1.4 Remarks to the Author: *"In order to bolster their conclusions regarding fenfluramine, the authors rely on comparisons with effects of dietary tryptophan depletion. Although some of these are indeed opposite in sign and therefore congruent with the present findings, the authors are often quite selective in their choice of which comparisons are made to favour their own findings. The same criticism applies to the use of the preclinical literature which is occasionally mis-cited. Hence, although this new study provides a useful new approach to the problem of 5-HT function, with some new findings, the conclusions are incremental rather than definitive."*

1.4 Author Response: Thank you for this comment. Constraints were placed on the breadth of the literature review; studies which we did not deem a close methodological fit were considered a lesser priority for discussion (e.g., pharmacological studies where the instrumental learning paradigm was of the reversal type). However, we have now made amendments to the manuscript to acknowledge the issue of methodological fit, as well as modifying the citations we have included.

We have reviewed all in-text citations to check for inappropriate inclusions. We have removed Montgomery *et al.*, and we have removed a further reference (Rygula *et al.*) in response to your later comment.

1.5 Remarks to the Author: *"Overall, the rationale of this new approach, reasonably enough, is to help clarify some of the apparent discrepancies found in previous studies, especially of effects of SSRIs. However, some of these 'discrepancies' probably result from such factors as duration of chronic treatment or differences in test paradigms for some*

cognitive constructs- and perhaps highlight the need for parametric exploration of these issues. it should be noted however that the present study also suffers from some of these same considerations, which should be acknowledged."

1.5 Author Response: Thank you for this comment. We agree that methodological differences could potentially explain discrepancies between some previous studies. However, it is important to note that some studies we reference in the manuscript use very similar paradigms. For example, Chamberlain *et al.* and Skandali *et al.* report inconsistent findings despite both measuring response inhibition using the Stop Signal Task at clinically relevant, acute, doses of SSRIs (citalopram/escitalopram). However, we agree that we should further highlight the importance of considering differences in study design/paradigms. We have now updated the manuscript, adding a paragraph within the Discussion (paragraph #10 of the section), to draw attention to the need to interpret the present findings within the context of the differences in methodology used in previous literature.

1.6 Remarks to the Author: *"The neural effects of sub-chronic fenfluramine are documented for rodents and not humans. This should perhaps be made clearer in the main text."*

1.6 Author Response: Thank you for this comment. We have made modifications to the introduction and discussion to make it clearer when we are referring to preclinical or human work.

1.7 Remarks to the Author: *"The rationale for the between-groups rather than cross-over study is not explained very clearly. The function of the baseline testing is also not made clear. Are there findings based on change scores? Was the study double blind?"*

1.7 Author Response: As the study design investigates subchronic administration of a drug or placebo (8 ± 1 days), a cross-over design would prove more logistically complicated (e.g., higher chance of attrition; carry-over and wash-out effects; reduced integrity of blinding). In addition, a cross-over study would require a within-subjects approach to analysis which is more vulnerable to biases introduced from individual variability (e.g., potential for learning between task sessions). A pre-post between-subjects design allows us to better isolate the treatment effect from such biases using baseline-adjusted ANCOVA (see responses to Reviewer #3 [3.2] comments for a more in-depth explanation, including the rationale for not using change score ANOVA). We have added more information about this design rationale to the manuscript. Finally, the study design was double blind; we have now clarified this in the updated manuscript.

1.8 Remarks to the Author: *"Behavioral tests. Is it completely accurate to call the verbal learning test one of "episodic memory" (in the sense of what, where and when?)."*

1.8 Author Response: Thank you for this comment. We have revised this in the manuscript now, referring to the verbal learning task as a measure of 'declarative memory'.

1.9 Remarks to the Author: *"Test of behavioral inhibition. It should perhaps be acknowledged that 'tests of behavioral inhibition' often in fact probe different processes. For the stop-signal task, another standard test of this construct, acute dietary tryptophan depletion does not affect performance in healthy volunteers (Clark et al Psychopharmacology 2005)- which is obviously not consistent with the conclusion of a "generalised" role for 5-HT in 'behavioral inhibition'. On the other hand, in a Go-NoGo test tryptophan depletion only induced disinhibition in the punishment condition (Crockett et al J. Neuroscience 2009). The latter is presumably consistent with the authors' hypothesis of selective valence effects of fenfluramine, although it is not cited. However, it should also be acknowledged that there are the limitations of using tryptophan depletion as a robust criterion of 5-HT deficiency, especially when the findings are anti-thetical to the observed findings here."*

1.9 Author Response: We agree that it is important to make the distinction between processes probed by differing behavioural inhibition paradigms. We note in our discussion that the Go/No-Go task measures action restraint, whereas Stop Signal tasks measure action cancellation (highlighted in yellow). In response to this comment, we have replaced the word 'generalised' within the text now with 'across emotional and neutral contexts', as our original intention was to convey that behavioural inhibition was increased across non-emotional and emotional contexts). We emphasise also that while the effects of TRP on behavioural inhibition in animals is relatively conclusive, we acknowledge in-text that this is only true to a "...lesser extent in humans", as cases such as Clark *et al.* 2005 demonstrate.

Thank you for mentioning the Crockett *et al.* 2009 paper. We have included this reference in the Introduction section of the manuscript now.

Finally, we have made one further addition to the discussion section to include mention of the potential limitations of TRP, citing Donkelaar *et al.*, 2011 and the subsequent letter response to the article by Crockett *et al.*, 2011.

1.10 Remarks to the Author: *"Test of instrumental learning. It was not quite clear to me how the best fitting model was derived. Which other competing alternatives were considered and how was the winning model chosen? Details should be provided."*

1.10 Author Response: Thank you for your comment. We provide a detailed answer to similar points raised by Reviewer #2 (**2.10**). In brief, the computational model for this task has been previously validated ⁷, with the best fitting model being the one used in the current study. Initially, we did consider two major forms of this model (non-reciprocal

and reciprocal updating types), and found that both models explained our behavioural data appropriately (see the updated Supplementary Results for further details).

1.11 Remarks to the Author: *“Not sure that Fig 4 and its accompanying elegant drift diffusion analysis in fact adds much to the overall understanding. Perhaps the legend needs to be more explicit.”*

1.11 Author Response: Thank you for this feedback. We agree that we initially did not make this very clear in the figure legend. As requested, we have revised parts of the figure legend to provide greater clarity on our rationale for undertaking DDM analysis now. These changes are highlighted in yellow in the figure legend.

1.12 Remarks to the Author: *“Was sex/gender considered as a factor for analysis? Perhaps merits some comment.”*

1.12 Author Response: Thank you for this thoughtful suggestion. In response, we have now included a gender analysis which includes gender as a factor within our ANCOVA modelling (see Supplementary Results; Supplementary Tables 7-9). Due to concerns about the sample size powering these analyses (particularly, a three-way interaction across group x gender x task condition), we ran these post-hoc analyses within separate ANCOVA models and have only included them in the Supplementary Information. We have noted that the inadequate statistical power limits the interpretability of these results.

There were no effects of gender on any of the primary effects reported within the main manuscript. However, we did observe an interaction effect (group x gender x task condition) on two AVLT outcomes (total repetitions and intrusions) which was not supported by group-level (drug vs placebo EMM) differences across task conditions. Since our primary finding on the AVLT was recalling total recall accuracy (where no gender effect was observed), we can (in part) rule out potential gender-related covariance for the primary treatment effects reported across the manuscript.

1.13 Remarks to the Author: *“The relevance of the cortisol test and findings was not made clear.”*

1.13 Author Response: We have amended to the manuscript to provide further detail on the rationale (Methods) and relevance of the findings (Results) of the cortisol analysis. As groups did not differ in cortisol levels during the initial drug administration window, we can rule out a significant modulation of HPA axis function from fenfluramine. While this has been observed at high dose fenfluramine⁸, it is possible the dose used in this study was insufficient to stimulate HPA axis response. However, we acknowledge that repeated measurement of cortisol throughout the whole study period would have been more informative. Hopefully this can be explored in a future research project.

1.14 Remarks to the Author: *“(i) Asymmetric effects of valence following tryptophan depletion have also been shown for model based versus model free processing (Worbe et al Mol Psychiat 2017).”*

1.14 Author Response: Thank you for this comment. It is difficult to directly compare the findings of certain past studies, including Worbe *et al.*, which use stay-shift behaviour as an index of learning rather than optimal choice behaviour. The RL task used in our study uses optimal choice behaviour as an index of model-free learning. Optimal choice behaviour reflects a more direct measure of learning behaviour compared to stay-shifts; stay-shift behaviour, while still interesting and informative, is more a measure of behavioural flexibility during learning^{9,10}. It is interesting that Worbe *et al.* found both the control and tryptophan depleted groups shifted to model-free strategy under reward conditions, although the control group was more mixed. However, the tryptophan-depleted group was the only instance where there was a unidirectional shift toward model-based strategy during punishment. Nevertheless, we emphasise caution on comparing our findings with other tasks in the literature without a close methodological fit now (see the revised Discussion section, line 431) and have included reference to Worbe *et al.* within the main manuscript.

1.15 Remarks to the Author: *“(ii) The claim that the preclinical, optogenetic literature supports the selective effects on aversive, as distinct from reward, processing is controversial to say the least. The paper by Rygula et al on my reading in fact seems to conclude no selective effects on aversive learning rate. I think these claims may need to be watered down....”*

1.15 Author Response: Thank you for this comment. We have removed the sentence at the start of the 4th paragraph in the discussion to tone down this claim. In addition, there was some initial hesitation about including Rygula *et al.* in the original manuscript due to the methodological approach used. The use of amygdala/OFC lesions to deplete 5-HT is far less targeted than optogenetic approaches, and is likely producing network/downstream effects in areas these neurons project through (*e.g.*, the hippocampus). Consequentially, ascertaining if the effects reported in Rygula *et al.* are related to aversive/reward processing or general impairments of cognition is difficult. As a result of these potential limitations, we have decided to omit reference to this study in the updated manuscript.

1.16 Remarks to the Author: *“(iii) The apparent conclusion that verbal memory retrieval was selectively affected over non-verbal memory is also apparently inconsistent with previously published effects of tryptophan depletion”*

1.16 Author Response: Thank you for raising this point. The effects of tryptophan depletion on memory function do appear to be weighted towards verbal memory, and in particular delayed recall. Indeed, in the largest systematic review of the cognitive effects of tryptophan depletion (Mendelsohn *et al.* ¹¹), the authors conclude: "...The most robust finding is that ATD impairs the consolidation of episodic memory for verbal information." The authors found inconclusive evidence of an effect on visuospatial learning and WM types.

Moreover, much of the tryptophan literature on verbal WM focuses on simple tasks (*e.g.*, digit span forwards) which are not sufficient measures of complex working memory such as the *n*-back. Indeed, the only study we are aware of involving tryptophan depletion and the verbal *n*-back task is Allen *et al.* ¹²; in their version of the *n*-back, there was no 3-back component which is where we found group differences in the current study. We have made modifications in-text (highlighted in yellow) to draw attention to insufficient evidence in this area, as this may complicate interpretation of this our findings on the verbal *n*-back task.

1.17 Remarks to the Author: (iv) Despite their literature review in the Supplementary Material, the authors fail to cite the important article by Luciana *et al.* (1998 Cerebral Cortex) on effects of fenfluramine on working memory in humans. An acute dose of 60mg impaired spatial working memory which perhaps should be commented upon.

1.17 Author Response: Thank you for highlighting the study by Luciana *et al.*. In the Supplementary Discussion we mention that aligning our current findings with past fenfluramine literature is challenging. The primary reason being those past studies of fenfluramine administered the agent at higher doses where there is greater potential for 5-HT depleting effects and/or neurotoxicity ^{13,14}. Indeed, there is preclinical evidence that higher dose fenfluramine reduces the density of 5-HT uptake sites and 5-HT/-HIAA levels, whereas low dose fenfluramine retains increased 5-HT/-HIAA levels without altering 5-HT neuron integrity ¹⁵. As a result, it is challenging to compare the current study, where fenfluramine was administered at 15mg B.I.D., to Luciana *et al.*, where a single dose of 60mg was administered. Indeed, the 5-HT depleting effects of high dose fenfluramine would account for the impairment of delayed spatial memory reported by Luciana *et al.*. Nonetheless, we have included Luciana *et al.* as a citation within the Supplementary Discussion in response to this suggestion.

1.18 Remarks to the Author: "10. Typo I think that serotonin is missing from the initial sentence of the Discussion (p15)"

1.18 Author Response: Thank you for highlighting this typo; we have now corrected this within the updated manuscript.

1.19 Remarks to the Author: *"SSRA- not sure this is yet fully justified as an abbreviation rather than referring simply to fenfluramine. This not yet an acknowledged category of psychoactive drugs."*

1.19 Author Response: Thank you for your comment. A primary aim of the paper is to understand effects of the SSRA mechanism on human behaviour. As we allude to in a previous response (1.17), the mechanism of fenfluramine may shift depending on its dose and enantiomeric composition/racemic form; as a result, at times in the paper we believe it is important to refer to it as an SSRA. We understand, however, that we must be more considered in its usage. We have now amended the 'Results' section of the manuscript to exclude any mention of the phrase 'SSRA', we have also reduced its usage in the introduction and discussion sections.

1.20 Remarks to the Author: *"Title of the ms. Seems a little pretentious, yet uninformative? "Selective effects of serotonin release on cognitive and aversive processing in humans" may be a more accurate and encompassing description of findings."*

1.20 Author Response: We appreciate your feedback on the title of the manuscript, and agree the title could be more informative. We have adapted the simplicity of your suggested title and have changed as follows:

"Direct serotonin release in humans selectively shapes aversive learning and inhibition"

We retain the 'direct' part of the original title in lieu of 'selective'. While the mechanism is selective to serotonin, we wish to avoid conflation with SSRIs for potential readership. Additionally, we omit the memory effects from the title to maintain brevity, particularly as these likely of less interest to readers. We hope you find this compromise appropriate.

Responses to Reviewer #2

2.1 Remarks to the Author: *"This is a very strong paper. It is concisely and clearly written, and well embedded within existing literature on serotonin's functional contributions. It reports the cognitive effects of a clinically highly innovative pharmacological probe (the selective serotonin releasing agent fenfluramine) which is mechanistically more potent and more precise than SSRIs, not suffering from the off-target dopaminergic effects or the acute-to-chronic mechanistic shift known to occur for SSRIs. As a result, it enables the authors to test more cleanly various predictions from longstanding theories of brain serotonin function. The paper neatly summarizes the results of a study in which 2 groups of healthy volunteers (27 vs 28) were given a low oral dose of fenfluramine for 7-9 days in a row, in a double-blind design. The results are fascinating, substantiating previous hypotheses about serotonin's role in behavioral inhibition, aversive slowing and memory enhancement. The paper represents a major contribution to the literature and will stir lots of new research, both basic science and clinically applied."*

2.1 Author Response: We would like to extend our gratitude to the reviewer for taking

the time to provide such insightful and constructive feedback on our work. We have done our best to address all the concerns you have raised within the manuscript. Please find our responses to each of your comments below.

2.2 Remarks to the Author: *"To clarify the key features of the study design (subchronic, 7-9 day oral administration, between-subjects, numbers of subjects per group!, pre- vs post-drug testing) at the beginning of the results section. Only after reading the methods section at the end of the paper did it become clear that the study was a between-group, pre/post-test design, involving intake of fenfluramine for multiple consecutive days."*

2.2 Author Response: Thank you for feedback, we have now included this information at the start of the Results section (highlighted in yellow).

2.3 Remarks to the Author: *"To more clearly justify the approach of using baseline performance as a regressor rather than adopting more generally employed/accepted repeated-measures ANOVAs or linear models. We were not sure we understand why the chosen approach would yield greater statistical efficiency and what the authors meant with the statement that this avoids conditional bias from baseline imbalance. This seems pertinent because the authors did choose to use the more classic linear modeling approach when analyzing cortisol and VAS and side effects."*

2.3 Author Response: This is an important point and agree it must be better justified within-text. This point was also raised by Reviewer #3 (3.2), we will refer to our response there for a more detailed justification of this decision. Based on this feedback, we have amended our justification within the 'Statistical Analysis' section of the main manuscript.

Regarding variability in modelling choices for cortisol, VAS and side effects data – due to the availability of more time points for these data (e.g., daily ratings), our linear modelling approach was altered to allow us to include all data points data with time as a regressor. This approach yields a more robust treatment effect estimate, however for consistency we also ran a separate baseline-adjusted ANCOVA on these data. We would have used this approach across all analysis if most study data were not limited to two time points (i.e., baseline and follow-up).

2.4 Remarks to the Author: *"Regardless of the above, I strongly recommend full reporting and visual presentation of all the data, including the baseline data."*

2.4 Author Response: Based on this feedback, we have now included visual presentation of all data, particularly baseline data, within the Supplementary Results. Please refer to the newly added Supplementary Figs. 5, 6, 7, 8, 9, 11, 12, 14, 16, 17, 18, 20, 21, 22, 23, and 24.

2.5 Remarks to the Author: *"To also plot, for full reporting purposes, drug effects on the win trials (not just the significant loss parameters) in the main text"*

2.5 Author Response: Thank you for this comment, we have now updated Fig 2C, D & F to plot data from the win trials alongside the loss trials.

2.6 Remarks to the Author: *"To report the effect of drug on response inhibition (% nogo) as a function of valence (or task interference: happy, fearful or control distractors) (analogous to the RT data analysis for this task). In the current version of the MS, fenfluramine is reported to increase response inhibition across all conditions. Can the stats be presented and data be plotted as a function of valence (or: 'task interference')? This will allow the reader to evaluate the degree to which the effect on response inhibition is indeed generalized or valence specific."*

2.6 Author Response: Based on this feedback, we have amended Fig 3B to plot the data across all three task conditions/valence.

2.7 Remarks to the Author: *"To clarify, justify, and possibly revise the interpretation of the response time effects in the go/nogo task in terms of improvements in impulse control (or enhanced action restraint). What aspect of this task or of the performance pattern makes the authors interpret the slowing of RTs as evidence for greater impulse control? I am not convinced the GO RTs represent impulsive choices in the first place. It seems equally possible that fenfluramine brought to the surface, rather than restrained impulsive behavior. After all, slowing on aversive trials has previously been conceptualized as an aversive Pavlovian (hardwired) inhibition 'impulse'. If the authors agree, then this would require some revision throughout the general discussion of the paper, which is referring to improved impulse control in various places. If they don't, then it would help to make more explicit the evidence for faster RTs on this task reflecting choice impulsivity."*

2.7 Author Response: Thank you for raising this important point. The interpretation of slowed response times for 'go' trials as a reduction in choice impulsivity might first seem unintuitive, particularly as slowed response time in cognitive tasks is usually seen as a less favourable outcome. However, slowing 'go' trial response times within the limited response window (400ms) represents an important shift in speed-accuracy trade-off, and a favourable optimisation in trial-by-trial task strategy. This is demonstrated by a positive correlational relationship between 'go' trial response times and response inhibition (or, 'no-go' trial accuracy) and cautious decision-making (log decision criterion [c]) within our task data:

Note: The relationship between 'go' RTs and 'no-go' accuracy has been reported previously in the literature ^{16,17}.

It is argued that a tendency toward premature, speeded responses without adequate signal processing constitutes the fundamental aspects of impulsive behaviour ^{18,19}. Accordingly, we note multiple publications which interpret response times to 'go' trials (in Go/No-Go tasks) as an index inhibition/impulsive responding ^{17,19-21}.

We have amended the manuscript to reflect these additional analyses, and included further rationale for this our interpretation of 'Go' RTs (highlighted in the Results section in yellow).

2.8 Remarks to the Author: *"Might the drug-related attenuation of loss sensitivity during learning reflect an indirect consequence of this increase in aversive Pavlovian inhibition? In this context, might it be informative to compute correlations between the effect on aversive GO RTs and the effect on loss learning."*

2.8 Author Response: Thank you once again for such considered feedback. As requested, we have computed the correlations between aversive 'Go' RTs and optimal choices during loss trials:

As you can see, the R value does not provide strong evidence for a relationship between these variables. Alongside this, the slowing of RTs in the task itself is not specific to 'Go' trials with aversive distractors, but is also observed during positive and control distractor trials (in addition to the generalised effect of the SSRA on 'no-go' trial accuracy). However, it is also important to consider the observation of an initial choice bias prior to evidence accumulation only during aversive distraction, however. The explanation with the greatest degree of parsimony, perhaps, is that 5-HT modulates behavioural inhibition irrespective of context, but that this effect is more pronounced in aversive contexts. We have included this reasoning in the 'Discussion' section of the manuscript:

"Potentially then, 5-HT performs an active role of limiting impulsive action across general contexts, but this is amplified in aversive contexts."

2.9 Remarks to the Author: *"To move a variety of key methods section from the supplement to the main manuscript. For example, the subjective VAS and side effect data are key, given that this is one of the first reports of this compound."*

2.9 Author Response: Thank you for this feedback. In response, we have moved what were previously Supplementary Tables 2 & 5 to the main manuscript (Tables 1 & 2, respectively). As you have requested, these tables contain summaries of analyses for self-reported data (including VAS and side effects).

2.10 Remarks to the Author: *"Furthermore, the reader would benefit from access to some more modeling details to evaluate the computational modeling procedures. What was the model (space)? Was any model comparison performed? Were simulations performed? Isn't there redundancy when modeling both outcome sensitivity and decision temperature? Why include the outcome sensitivity parameter? Is this the same choice stochasticity parameter that was previously found to be modulated by serotonin"*

manipulations (e.g. acute tryptophan depletion by Ben Seymour et al)?"

2.10 Author Response: We appreciate these considered thoughts on our approach to computational modelling. For each approach (RL and DDM), we have now included further details on the modelling space, parameter recovery and simulation procedures within the Supplementary Methods.

Both RL and DDM modelling configurations for each task are based on previously published and validated approaches^{7,22,23}. While model alteration and subsequent model comparison were considered, it was decided that use of a previously validated approach would offer more benefits (*e.g.*, avoidance of intra-study bias generated from reparameterising the model for a specific set of data) and demonstrates the reliability and generalisability of the model^{24,25}. Additionally, as shown in Supplementary Figs. 1 and 4, simulation from model-fitted parameters result in a close match to patterns observed in the real task data.

Nevertheless, we did consider an alternative version of the RL model without reciprocal value updating (*i.e.*, values were only updated for the chosen stimulus within each pair, not the unchosen one). This version of the model performs equally well in explaining the patterns in our behavioural data; we have included this analysis in the Supplementary Results now. We ultimately decided to retain the reciprocal-updating version of the model, however, as this resulted in the best performance within model validation by Halahakoon et al.⁷.

The original manuscript did not contain precise descriptions of our RL modelling procedures which may be contributing to confusion. We have amended the Results/Methods sections of the main manuscript and Supplementary Methods to provide greater clarity on the modelling procedure.

Regarding redundancy in outcome sensitivity and decision temperature parameters; these parameters are estimated in two separate models. While they are designed to capture distinct aspects of the decision-making process, they mathematically converge on the same point²⁶. As a result, we must use prior evidence from the serotonergic system to decide the most appropriate parameter to describe patterns in the behavioural data. Neurobiological evidence correlates neural burst firing in amygdalae 5-HT neurons to negative prediction errors during punishment, reflecting a neural signature of outcome sensitivity^{27,28}. In contrast, there is no literature which suggests a link between 5-HT neuronal activity and choice stochasticity during model-free learning. This prior evidence guided the selection of the computational model for our primary analyses, where our model contains only the outcome sensitivity parameter and learning rate. For the sake of transparency, we report the findings from the decision temperature model and demonstrate their mathematical redundancy (Supplementary Fig 1).

Regarding the choice stochasticity parameter used in Seymour *et al.*²⁹, this is not quite an equivalent to the parameter used in our paper as the underlying computational model differs in how this is calculated. Within their study, punishment and reward were possible outcomes on every trial, whereas in our study there were two trial types (win vs no change; loss vs no change), allowing us to model outcome sensitivity to reward and punishment separately. Within Seymour *et al.*²⁹, they report reduced reward sensitivity in tryptophan depleted individuals, however, as both punishment and reward were learned concurrently, it is probable that reduced sensitivity to reward is a direct consequence of bias toward avoiding punishment. Nevertheless, as we noted in an earlier response to Reviewer #1 (1.5), it is important to stress the underlying differences in task design and modelling approach that can contribute to difficulties in making comparisons with past literature.

2.11 Remarks to the Author: *“To clarify whether a power calculation was performed to justify the sample sizes.”*

2.11 Author Response: This is an important consideration; a power calculation was performed and has now been inserted in-text in the Statistical Analysis section:

“A priori sample size calculation (Power [1 - β error probability]: 80%) required a sample size of $N=52$ (26 per group) to undertake two-tailed between-groups analyses.”

2.12 Remarks to the Author: *“I would suggest breaking down the group x task interaction for GO response times in a way that makes the nature of the interaction more*

obvious. Was there perhaps a simple main effect of valence (aversive vs control) for drug, but not placebo? And what about positive vs control?"

2.12 Author Response: We have now broken down the group x task interaction further as suggested. Indeed, we observed a main effect of valence (aversive vs control) within the SSRA group but not the placebo group. We have summarised these results in Supplementary Table 8:

Supplementary Table 8. Valence-specific contrasts for 'Go' trial response times in the Affective Interference Go/No-Go Task

	F-Statistic ^a	Degrees of Freedom	P-Value
SSRA Group (n=24)			
Aversive vs Control	4.87	1,22	0.04
Happy vs Control	3.05	1,22	0.10
Placebo Group (n=26)			
Aversive vs Control	1.95	1,24	0.28
Happy vs Control	1.00	1,24	0.33

^a via baseline-adjusted ANCOVA type II modelling.

We have also amended the main manuscript text to include these results.

2.13 Remarks to the Author: *"Page 15, line 273: is the word '5-HT' missing after 'synaptic'?"*; *"Page 16, line 308: is the word '-projecting' missing after '5-HT'?"*

2.13 Author Response: Thank you for spotting these typos, we have corrected these in the updated manuscript.

Responses to Reviewer #3

3.1 Remarks to the Author: *"In their manuscript „Direct serotonin release in humans shapes decision computations within aversive environments“, Colwell et al. use behavioural testing, computational modelling and pharmacological neuromodulation to examine the impact of repeated oral administration of a serotonergic agent (the SSRA fenfluramine) on cognitive measures including learning, behavioural inhibition and memory. The authors found that serotonin reduces sensitivity to aversive outcomes during learning, enhances behavioural inhibition, and modulates memory for verbal information. Overall, the study is well-executed, the manuscript is well-written, and the results are important. I believe, the findings advance the canon of knowledge in the field. However, I think the following points could be clarified and improved in the manuscript."*

3.1 Author Response: We thank the reviewer for taking the time to read through our manuscript and providing considered feedback. We have addressed all your comments below and updated the manuscript based on your suggestions.

3.2 Remarks to the Author: *"1) As far as I understand, the authors tested subjects twice, once pre-treatment, and once after/during treatment. They use baseline score as regressor in ANOCVA. This is critical. It seems to me the authors have thought about this carefully. However, I feel, it'd be important to back this approach up with some data. I think, the least the authors should do is show the baseline data. The authors state by citing references that this is in their opinion the best way to analyse the data. However, they refer to clinical trials. This is not trivial, given cognitive neuroscience trials using cognitive tasks (as in the current study), feature characteristics that clinical trials may lack, e.g., learning effects when testing subjects repeatedly. Thus, referring to clinical trials may not suffice when making this important analytical decision. Relatedly, what is the advantage of having a baseline session in the first place, or in other words, what was the rationale for the study design? Because familiarising the participants with the task can also be done in other ways, e.g., training before the actual task session. Thus, it'd be important to know more about the decision process re. the study design and analytical approach? Further, this analytical approach should be more clearly mentioned in the results section (similar to the supplement, where the authors state "baseline-adjusted ANCOVA")."*

3.2 Author Response: Thank you for highlighting this important point. While baseline-adjusted ANCOVA is frequently employed in the context of clinical trials, its application is less conventional in typical cognitive neuroscience research. Nevertheless, it is essential to recognize that the current study exhibits substantial similarities with standard clinical trial designs, notably randomised group allocation and placebo control. Moreover, this study was conceived as a potential first step within an experimental medicine pipeline, leading to a potential follow-up study in a clinical population where a baseline-adjusted ANCOVA approach would be expected. Indeed, in direct comparison ANOVA of change, the baseline-adjusted ANCOVA yielded more statistical power in a clinical study of depressed individuals ³⁰.

A further justification for baseline-adjusted ANCOVA is that it isolates the treatment effect from potential sources of biases, including learning and practice effects. If we assume that these effects are a unidirectional source of variability (typically positive), we can better isolate the effect of the SSRA by regressing against baseline performance ^{30,31}. As an example from our dataset, consider these data from the PILT task which shows the relationship between optimal choices during loss trials at baseline and follow-up:

As you can see, the positive relationship between performance at baseline and follow-up reflect the degree to which unseen sources of inter-participant variability at baseline (e.g., learning potential) may have contributed to performance at follow-up. By adjusting performance based on the participant's starting point via ANCOVA, we can better isolate the treatment effect from these sources of variability. In comparison, ANOVA of change (pre-post difference) assumes that changes over time are attributable solely to treatment effect and are not related to other sources of variability (i.e., learning effects which differ between individuals).

We agree that it is important to see baseline data and have now included this in our Supplementary Results. In most cases each group was balanced at baseline. However, even when baseline variables are balanced, ANCOVA still offers more precise effect estimate over ANOVA of change ^{32,33}.

We note that while the approach is less widely used outside of clinical trials, studies similar in design have implemented it (e.g., a pre-post study concerning tDCS effects on verbal *n*-back task performance in *npj Science of Learning* ³⁴). There are other examples of baseline-adjusted ANCOVA yielding greater power vs ANOVA of change outside of clinical trials, such as in educational psychology research ³⁵.

We have amended the manuscript to provide a more precise rationale for our approach. In addition, we state at the start of the Results section that all primary analyses involved baseline-adjusted ANCOVA modelling.

3.3 Remarks to the Author: "2) *The authors mention that baseline performance served as*

a regressor and participant as a random effect "where appropriate" (cf. Supplementary Material). Here, I think, it is important to clarify what "where appropriate" means?"

3.3 Author Response: We agree that this was not specific enough, and have amended the manuscript accordingly:

"...with baseline performance serving as a regressor and participant as a random effect due to nested multilevel data structures (e.g., multiple task conditions)."

3.4 Remarks to the Author: *"3) FWE correction is mentioned in the methods section. However, where was this applied in the actual analysis? Given the many analyses, and the lack of clearly defined hypotheses for many cognitive tasks reported, this seems important."*

3.4 Author Response: Thank you for your feedback. We applied the Bonferroni-Holm procedure to estimated marginal means tests, correcting for Family-Wise Error within individual tasks (across task conditions). We have now updated the manuscript to include this detail.

While we considered applying false discovery rate correction cross-analyses (e.g., Benjamini-Hochberg), we anticipated challenges due to the inherent heterogeneity of the dataset. Data properties such as modality, complex dependencies, structure, and temporal dimensions makes algorithmic implementation complex and raised concerns about the potential for Type II errors. As a result, we did not to use FDR correction cross-analyses.

3.5 Remarks to the Author: *"4) In my view, there is missing detail on the methods of the learning task. For instance: What are the number of trials, number of stimulus pairs, number of blocks in the task?"*

3.5 Author Response: We apologise that these details were missing and have updated the manuscript to include them within the task description in-text and within the figure legend in the Results section.

3.6 Remarks to the Author: *"5) Further, if I am not mistaken, the authors mention they only analysed the last 40 trials of each block (cf. Supplementary Material). I assume, performance then is stable (which makes sense when analysing choice behaviour), however, I also assume a major bulk of learning is taking place before that? Given the authors use a RL learning model, I don't fully understand why the authors take this approach, and I wonder whether the authors can elaborate on this approach a bit more?"*

3.6 Author Response: We understand that the way we have described our approach in the original manuscript was potentially confusing. All task trials were fitted within the RL

modelling procedure. We analysed only the last 40 trials for optimal choice behaviour (non-model data), consistent with previous work using this task, as this is where the learning plateau typically occurs^{36,37}. We have amended the manuscript to provide greater clarity on this approach now.

3.7 Remarks to the Author: *"The authors (in the Supplementary Material) mention "total gained" as a performance measure in the methods. However, I don't find the results in the results section. Given performance is different between drug groups (worse, as it seems, for the SSRA), it'd be interesting to have some kind of measure of overall performance (which may, or may not be, the authors can clarify here, total points gained)?"*

3.7 Author Response: We have now included an analysis on the total amount earned within the Supplementary Results, including a numerical comparison with the baseline data. We observed no effect of group allocation on total money earned at follow-up. As both groups performed equally well in gain trials, it is likely that the decrease in optimal choices during loss was not sufficient to drive a significant effect on total earned.

3.8 Remarks to the Author: *"7) The authors mention that the SSRA reduced the number of optimal choices during loss trials only (cf. Fig. 2D). As mentioned above (cf. comment #3), I wonder whether the authors can elaborate on this result a bit more, as it seems this relates to worse performance in the task? If so, I believe this is worth mentioning and discussing, also, but not only, because performance on other tasks (cf. below) is improved."*

3.8 Author Response: We understand that reduced optimal choices for loss trials in the SSRA group might appear as worse performance. While this could be the case, it is unlikely this reflects impaired cognition as such. Indeed, this effect was specific to loss learning. Further, the SSRA group showed improved verbal learning and working memory ability in other tasks. In our RL modelling, we can see that the effect itself was specifically driven by reduced sensitivity to aversive outcomes rather than reductions in learning rate. In the real world, a reduced sensitivity to loss could be considered adaptive depending on circumstance.

A further point worth mentioning: if we consider objective performance on the task as a measure of the total money earned, then there was no significant difference between groups (as mentioned in an earlier response).

We have now amended the 'Discussion' to include a paragraph where we discuss this issue.

3.9 Remarks to the Author: *"8) Supplementary Material, p3, line 125-126. The description of the outcome coding is not clear to me. Shouldn't it be "0" in loss trials instead of "no change", and "-1" in loss trials? The coding is critical when analysing and interpreting valence (positive/negative outcomes) effects."*

3.9 Author Response: This is an oversight on our part, and we appreciate you noticing this. We have amended the text now:

"while R_t refers to the positive (coded as '1' for win in win trials, and '0' for no change in loss trials) or negative (coded as '0' for no change in win trials, and '-1' for loss in loss trials) outcome observed."

3.10 Remarks to the Author: *"9) "The unchosen option was not kept constant but rather updated based on the reciprocal outcome", cf. Supplementary Material p4. Maybe the authors can elaborate on this approach for the reader to understand? If task design is explained more detailed (cf. above), then this should be much clearer."*

3.10 Author Response: In the previous version of the RL, only the value expectations for the chosen symbol each updated cycle, leaving the unchosen option unchanged. Previous model validation ⁷, however, shows that a kernel which reciprocally updates expectations for the unchosen option (based on counterfactual outcome) results in slightly better model fit. Therefore, we opted to use the reciprocal-updating model. However, as we now show in the updated Supplementary Methods/Results, the model without reciprocal-updating results in the same behavioural differences across groups.

We have amended the description of reciprocal updating in the model. To provide extra clarity for readers, we have also added a paragraph describing the model without reciprocal-updating, and provided a rationale for why we chose the reciprocal-updating model.

3.11 Remarks to the Author: *"10) The authors mention (and show, cf. Supplementary Material) a "perfect correlation" between the decision temperature and outcome sensitivity parameters. This makes me wonder: Why did the authors not compare different models with different sets of parameters? It is unclear to me why the authors opt for this model, given it seems that different parameters are capturing the same effects/processes, and are, as it seems, redundant. Such problems can be addressed with, for instance, model comparison, of different models with different sets of model parameters. Given the authors make strong claims based on the "computational" results, this seems critical to me. Given also that the model-free results are convincing and strong, there is no doubt that that the modelling results can reveal important mechanistic insights, however, as is, I find the interpretation of the modelling results rather difficult."*

3.11 Author Response: Thank you for this comment. As it shares overlaps with concerns shared by Reviewer #2 (2.10), we refer to our detailed response provided there. In brief, we derive outcome sensitivity and decision temperature from separate models, however, there is no way to computationally distinguish between these parameters (the scatterplot demonstrates this effect through a direct comparison of the fixed parameters across

each model). As a result, we must use prior evidence from the serotonergic system to decide the most appropriate parameter to describe patterns in the behavioural data.

We have now amended the manuscript and Supplementary Methods to address these key issues.

3.12 Remarks to the Author: *"11) Related to the prior comment: what is the difference between the learning rates and outcome sensitivity parameters, and what is the rationale for having both parameters, and on top of that, having them separate for win/loss trials? What is the rationale for having so many parameters, are different parameters needed for win/loss trials? I believe that such questions can be addressed by model comparisons, for instance, having different models that have separate learning rates for win/loss, and shared learning rates, etc. are needed."*

3.12 Author Response: Learning rate determines the slope of the learning curve, while outcome sensitivity determines the asymptote⁷. While these parameters are separable, decision temperature and outcome sensitivity are not (as covered in our previous response). Regarding the choice to have separate win and loss parameters is supported by patterns observed in non-model optimal choice behaviour, specifically the group x valence interaction. If loss and win trials were fed into the same model, we would not be able to detect a valence-specific effect (as observed within the non-model behavioural data). We have amended the model description in the Supplementary Methods to include this information.

3.13 Remarks to the Author: *"12) Related to the prior comments: I believe, the authors need to present simulated data (from the winning models) and model recovery results (for the parameters) as is common practice in cognitive computational research to make sure the models are actually capturing what the authors claim they do."*

3.13 Author Response: Thank you for this comment. In response to this feedback, we have provided details on simulated data and parameter recovery within the Supplementary Methods now.

3.14 Remarks to the Author: *"13) The results for the behavioural inhibition task – if my interpretation is correct – show that inhibitory processes (no-go) are most impacted by the SSRA when aversive distractors are present. It'd be interesting to have some interpretation of this result, given it shows that subjects were more sensitive to aversive stimuli, which seems to contrast with the learning results, where subjects were less sensitive to aversive stimuli?"*

3.14 Author Response: This is an interesting point; there are a few results to consider when drawing parallels to the effect on loss learning. Importantly, response times for 'go'

trials were slowest in the SSRA group, particularly during aversive interference. Slowing response times is an adaptive optimisation in task strategy (see our response to Reviewer #2 [2.7] for further details). Beyond this, SSRA allocation resulted in a stronger bias toward the impulse control ('no-go') boundary during aversive interference (via DDM). Taken together, these results suggest that aversive interference has less detrimental effects on task optimisation in the SSRA group, which is consistent with a reduction in loss sensitivity.

We have amended the manuscript now to include a sentence which considers the effects of the SSRA on both tasks.

3.15 Remarks to the Author: "14) Related to the prior comment: is it fair to say that SSRA subjects performed better in the behavioural inhibition task? And is that different to the learning results (cf. comments #3 and #4). This is by no means a methodological critique, it'd just be interesting to hear about the authors' interpretation of these results.

3.15 Author Response: It is certainly fair to say the SSRA group performed better in the behavioural inhibition task. Response accuracy for no-go trials was higher, which was facilitated by a slowing of response times; a slowing of response time within the limited response window (400ms) can be considered an optimisation in task behaviour, as it is correlated with improved accuracy for no-go trials (see our response to Reviewer #2 [2.7] for further details). Regarding the learning results, reinforcement learning and behavioural inhibition tasks capture broadly distinct psychological phenomena and are modulated by different neural pathways^{28,38}. In other words, these results are not necessarily related to each other, though they share a valence overlap. However, as we note in our previous response, we do not consider these results inconsistent with each other.

Additionally, we have updated our description of the response inhibition task in the Fig 3 legend now to include number of task blocks and trials per block.

3.16 Remarks to the Author: "15) Did I understand correctly that subjects took the drug repeatedly (daily) for multiple days? If so, I think, this is (one of the many) strength of this study, and I think, the authors could more prominently inform the reader about this in the abstract, introduction, and results (as it is, it is "hidden" in the methods only)."

3.16 Author Response: Thank you for this suggestion. We have amended the manuscript now so that the abstract, introduction and results contain mention of the subchronic nature of SSRA administration.

3.17 Remarks to the Author: "16) Similar to my prior comment, I believe, the authors should consider placing the results of the placebo/drug allocation guess in the results

section rather than put it in the methods only. I think, this is an important result that should be mentioned in the main manuscript, partly (not only) because it addresses an often-raised criticism of clinical trials in psychiatry on, e.g., antidepressant drugs. However, this is up for the authors to decide, no need to please the reviewer here, but rather a suggestion.”

3.17 Author Response: We appreciate this suggestion and have moved these findings to Results section in the main manuscript now.

3.18 Remarks to the Author: “17) In the discussion and conclusion section, the authors mention “more neutral contexts”. I think, it would help if the authors clarified this, as it is not clear to me what this is supposed to mean.”

3.18 Author Response: We understand that this is quite confusing. We wanted to avoid using the term “neutral context” in the main manuscript without acknowledging most (if not all) contexts have an inherent valence, albeit often it is implicit. However, we have now added a qualifier for the first time we write “neutral contexts” within the main text, which is hopefully less confusing to readers:

“...underlining its critical role in guiding decision-making across aversive and more neutral contexts (i.e., where valence is not explicitly manipulated).”

3.19 Remarks to the Author: “18) Given the robust and important findings, I think, it is justifiable for the authors to at least – albeit cautiously – discuss their findings with re. to implications for treatment of psychiatric disorders. It’d be nice to hear some of the author’s ideas about how these results relate to their own prior extensive work (e.g., on SSRIs) and discuss clinical implications.”

3.19 Author Response: Thank you for this suggestion. It is indeed tempting to consider the potential clinical implications of these findings, however, we have been cautious about including a detailed discussion of this in the manuscript given it is rather speculative. However, we do consider some of the broader clinical implications of this work in the conclusion:

“Given the prominence of impaired cognition and aversive/negative emotional biases as transdiagnostic targets within psychiatry (e.g., unipolar and bipolar depression; schizophrenia)^{39,40}, investigating the therapeutic potential of the SSRA in clinical populations may be worthwhile. Such investigations may allow greater targeting of specific neurocognitive mechanisms across disorders in the absence of widespread, and often unwanted, effects including emotional blunting.”

We have also added an additional sentence earlier in the discussion where we discuss the selective effects of the SSRA on reinforcement learning:

"It would be worthwhile to investigate if the selective effect of the SSRA on aversive processing could prove advantageous for the treatment of depression without exacerbating features of anhedonia."

3.20 Remarks to the Author: "19) Supplementary fig. 2/3 is wrongly referenced in supplementary methods, e.g., p6 line 223/227; the respective figures do not show the referenced results. So there seems to be a mix-up here."

3.20 Author Response: We have rectified this issue in the updated manuscript – thank you for pointing this out.

3.21 Remarks to the Author: "20) p5, line 99: *"historically considered historically core functions of serotonin"*. Consider revising this sentence."

3.21 Author Response: We have omitted the redundant "historically" now – once again, thank you for pointing this out.

3.22 Remarks to the Author: "21) *Supplementary Material, p5, line 210: Results "bordered" significance. Further, the reader is not informed about the direction of results. I think that showing "bordered" (not corrected for multiple comparisons?), non-significant results sort of renders using alpha levels of no avail.*"

3.22 Author Response: We agree that this invalidates setting alpha levels, and we have removed mention of the word "bordered" now.

Additional Changes

- We have amended the manuscript to explicitly state that we are reporting on all primary outcome measures within the clinicaltrials.gov (NCT05026398) study registration.
- We have made further formatting changes (*e.g.*, proper labelling of figure titles; removal of subheadings in the discussion) to better align with formatting guidelines of the journal.
- Minor issues were fixed in the DDM and *n*-back preprocessing scripts, resulting in small changes in the reported statistics. The statistical relationships are generally consistent with those reported previously. However, we noticed a new statistically significant effect of group on the DDM parameter 'non-decision time', however this was not supported by a significant group difference via EMM testing.

References for author responses

1. Kapur, S., Meyer, J., Wilson, A. A., Houle, S. & Brown, G. M. Modulation of cortical neuronal activity by a serotonergic agent: a PET study in humans. *Brain Res* **646**, 292–294 (1994).
2. Meyer, J. H. *et al.* Neuromodulation of frontal and temporal cortex by intravenous d-fenfluramine: an [¹⁵O]H₂O PET study in humans. *Neurosci Lett* **207**, 25–28 (1996).
3. Quednow, B. B. *et al.* Assessment of serotonin release capacity in the human brain using dexfenfluramine challenge and [¹⁸F]altanserin positron emission tomography. *Neuroimage* **59**, 3922–3932 (2012).
4. Ikoma, Y. *et al.* Measurement of changes in endogenous serotonin level by positron emission tomography with [¹⁸F] altanserin. *Ann Nucl Med* **35**, 955–965 (2021).
5. Dileep Kumar, J. S. & John Mann, J. PET tracers for serotonin receptors and their applications. *Central Nervous System Agents in Medicinal Chemistry (Formerly Current Medicinal Chemistry-Central Nervous System Agents)* **14**, 96–112 (2014).
6. Mach, R. H. & Luedtke, R. R. Challenges in the development of dopamine D₂-and D₃-selective radiotracers for PET imaging studies. *J Labelled Comp Radiopharm* **61**, 291–298 (2018).
7. Halahakoon, D. C. *et al.* Pramipexole Enhances Reward Learning by Preserving Value Estimates. *Biol Psychiatry* (2023) doi:<https://doi.org/10.1016/j.biopsych.2023.05.023>.
8. Schürmeyer, T. H., Brademann, G. & von zur Mühlen, A. Effect of fenfluramine on episodic ACTH and cortisol secretion. *Clin Endocrinol (Oxf)* **45**, 39–45 (1996).
9. Evenden, J. L. & Robbins, T. W. Dissociable effects of d-amphetamine, chlordiazepoxide and α-flupenthixol on choice and rate measures of reinforcement in the rat. *Psychopharmacology (Berl)* **79**, 180–186 (1983).
10. Bartolo, R. & Averbeck, B. B. Prefrontal cortex predicts state switches during reversal learning. *Neuron* **106**, 1044–1054 (2020).
11. Mendelsohn, D., Riedel, W. J. & Sambeth, A. Effects of acute tryptophan depletion on memory, attention and executive functions: A systematic review. *Neurosci Biobehav Rev* **33**, 926–952 (2009).
12. Allen, P. P. *et al.* Effect of acute tryptophan depletion on pre-frontal engagement. *Psychopharmacology (Berl)* **187**, 486–497 (2006).
13. Baumann, M. H., Ayestas, M. A. & Rothman, R. B. Functional consequences of central serotonin depletion produced by repeated fenfluramine administration in rats. *Journal of Neuroscience* **18**, 9069–9077 (1998).
14. Kalia, M. Reversible, short-lasting, and dose-dependent effect of (+)-fenfluramine on neocortical serotonergic axons. *Brain Res* **548**, 111–125 (1991).
15. Zaczek, R. *et al.* Effects of repeated fenfluramine administration on indices of monoamine function in rat brain: pharmacokinetic, dose response, regional specificity

- and time course data. *Journal of Pharmacology and Experimental Therapeutics* **253**, 104–112 (1990).
16. Sánchez-Kuhn, A. *et al.* Go/No-Go task performance predicts differences in compulsivity but not in impulsivity personality traits. *Psychiatry Res* **257**, 270–275 (2017).
 17. Bezdjian, S., Baker, L. A., Lozano, D. I. & Raine, A. Assessing inattention and impulsivity in children during the Go/NoGo task. *British Journal of Developmental Psychology* **27**, 365–383 (2009).
 18. Dalley, J. W., Everitt, B. J. & Robbins, T. W. Impulsivity, compulsivity, and top-down cognitive control. *Neuron* **69**, 680–694 (2011).
 19. Campanella, S. *et al.* Short-term impact of tDCS over the right inferior frontal cortex on impulsive responses in a go/no-go task. *Clin EEG Neurosci* **49**, 398–406 (2018).
 20. Ruchow, M. *et al.* Impulsiveness and ERP components in a Go/Nogo task. *J Neural Transm* **115**, 909–915 (2008).
 21. Zhao, X., Qian, W., Fu, L. & Maes, J. H. R. Deficits in go/no-go task performance in male undergraduate high-risk alcohol users are driven by speeded responding to go stimuli. *Am J Drug Alcohol Abuse* **43**, 656–663 (2017).
 22. de Gee, J. W. *et al.* Pupil-linked phasic arousal predicts a reduction of choice bias across species and decision domains. *Elife* **9**, e54014 (2020).
 23. Ratcliff, R., Huang-Pollock, C. & McKoon, G. Modeling individual differences in the go/no-go task with a diffusion model. *Decision* **5**, 42 (2018).
 24. Miłkowski, M., Hensel, W. M. & Hohol, M. Replicability or reproducibility? On the replication crisis in computational neuroscience and sharing only relevant detail. *J Comput Neurosci* **45**, 163–172 (2018).
 25. Forster, M. R. Key concepts in model selection: Performance and generalizability. *J Math Psychol* **44**, 205–231 (2000).
 26. Browning, M., Paulus, M. & Huys, Q. J. M. What is computational psychiatry good for? *Biol Psychiatry* **93**, 658–660 (2023).
 27. McHugh, S. B. *et al.* Aversive prediction error signals in the amygdala. *Journal of Neuroscience* **34**, 9024–9033 (2014).
 28. Sengupta, A. & Holmes, A. A Discrete Dorsal Raphe to Basal Amygdala 5-HT Circuit Calibrates Aversive Memory. *Neuron* **103**, 489–505.e7 (2019).
 29. Seymour, B., Daw, N. D., Roiser, J. P., Dayan, P. & Dolan, R. Serotonin selectively modulates reward value in human decision-making. *Journal of Neuroscience* **32**, 5833–5842 (2012).
 30. Van Breukelen, G. J. P. ANCOVA versus change from baseline had more power in randomized studies and more bias in nonrandomized studies. *J Clin Epidemiol* **59**, 920–925 (2006).

31. Wright, D. B. & Osborne, J. E. Dissociation, cognitive failures, and working memory. *Am J Psychol* **118**, 103–114 (2005).
32. Egbewale, B. E., Lewis, M. & Sim, J. Bias, precision and statistical power of analysis of covariance in the analysis of randomized trials with baseline imbalance: a simulation study. *BMC Med Res Methodol* **14**, 1–12 (2014).
33. Wei, L. & Zhang, J. Analysis of data with imbalance in the baseline outcome variable for randomized clinical trials. *Drug Inf J* **35**, 1201–1214 (2001).
34. Ke, Y., Liu, S., Chen, L., Wang, X. & Ming, D. Lasting enhancements in neural efficiency by multi-session transcranial direct current stimulation during working memory training. *NPJ Sci Learn* **8**, 48 (2023).
35. Petscher, Y. & Schatschneider, C. A simulation study on the performance of the simple difference and covariance-adjusted scores in randomized experimental designs. *J Educ Meas* **48**, 31–43 (2011).
36. Pessiglione, M., Seymour, B., Flandin, G., Dolan, R. J. & Frith, C. D. Dopamine-dependent prediction errors underpin reward-seeking behaviour in humans. *Nature* **442**, 1042–1045 (2006).
37. Gillespie, A. L., Wigg, C., Van Assche, I., Murphy, S. E. & Harmer, C. J. Associations Between Statin Use and Negative Affective Bias During COVID-19: An Observational, Longitudinal UK Study Investigating Depression Vulnerability. *Biol Psychiatry* **92**, 543–551 (2022).
38. Rae, C. L., Hughes, L. E., Anderson, M. C. & Rowe, J. B. The prefrontal cortex achieves inhibitory control by facilitating subcortical motor pathway connectivity. *Journal of neuroscience* **35**, 786–794 (2015).
39. Colwell, M. J. *et al.* Pharmacological targeting of cognitive impairment in depression: recent developments and challenges in human clinical research. *Transl Psychiatry* **12**, 484 (2022).
40. Vrieze, E. *et al.* Reduced reward learning predicts outcome in major depressive disorder. *Biol Psychiatry* **73**, 639–645 (2013).

REVIEWERS' COMMENTS

Reviewer #1 (Remarks to the Author):

The ms is much improved by the revisions. The renewed study of fenfluramine in humans is certainly worthwhile. However, there are some residual issues:

1. The authors are rather cavalier in their interpretation of the optogenetics literature. There are several articles (including more recent ones than their cited report by McDevitt et al) reporting that serotonin neurons are activated by reward as well as aversive stimulation; I list a few of these articles below:

Bromberg-Martin ES et al (2010) Journal of Neuroscience 30, 6252-6272 (2010)

Cohen et al eLife 4,; e06346 (2015)

Matias et al eLife 6;e20552 (2017)

Zhong et al J Neuroscience 37, 8863-8875 (2017)

Miyazaki et al Nature Communications 9(1), 1000 (2018)

Paquelet GE et al Neuron 111 2664-2679 (2022).

I think the authors should consider this literature much more carefully when interpreting their results.

2. Incidentally, the omission of Rygula et al on the methodological grounds stated in the rebuttal appears curious and illogical. Optogenetic manipulation of 5-HT neurons will almost certainly also produce network/downstream effects in areas these neurons project through. Optogenetic manipulations have not so far to my knowledge reproduced effects of regionally and neurochemically selective effects of serotonin depletion, especially in primates. To attribute the selective effects shown by reinforcement theory modelling in that study to "general cognitive impairments" sounds like hand-waving. In fact, the results of Rygula et al support the general view of the optogenetics literature that 5-HT is involved in both reward and punishment processing, (specifically in its anticipation), apparently contradicting the present report. I think the authors should concede that their report will be somewhat controversial in relation to the preclinical literature.

3. Minor. The revision on lines 95-97 is missing some words.

Reviewer #2 (Remarks to the Author):

The revisions of the text and the additional analyses as well as the greater clarity of the analytical (baseline-correction and computational learning modeling) procedures were much appreciated. It is clear the authors have exerted great effort and caution to now fully report all the analyses, data, simulations and parameter recovery. I am now convinced that the various choices are justified.

Typo Fig 2 legend: per of symbols should be pair of symbols. Fig 2F one 'in the' too many

Reviewer #3 (Remarks to the Author):

The authors successfully addressed my concerns.

Point-by-point responses to reviewer comments (NCOMMS-23-46547A)

Responses to Reviewer #1

1.1 Remarks to the Author: *"The ms is much improved by the revisions. The renewed study of fenfluramine in humans is certainly worthwhile. However, there are some residual issues:"*

1.1 Author Response: We would like to thank the reviewer for their constructive feedback throughout the review process. We now further amended the manuscript to address these comments you have raised below.

1.2 Remarks to the Author: *"1. The authors are rather cavalier in their interpretation of the optogenetics literature. There are several articles (including more recent ones than their cited report by McDevitt et al) reporting that serotonin neurons are activated by reward as well as aversive stimulation; I list a few of these articles below:*

Bromberg-Martin ES et al (2010) Journal of Neuroscience 30, 6252-6272 (2010)

Cohen et al eLife 4,; e06346 (2015)

Matias et al eLife 6;e20552 (2017)

Zhong et al J Neuroscience 37, 8863-8875 (2017)

Miyazaki et al Nature Communications 9(1), 1000 (2018)

Paquelet GE et al Neuron 111 2664-2679 (2022).

I think the authors should consider this literature much more carefully when interpreting their results."

1.2 Author Response: Thank you for raising this important issue. We agree that the optogenetic labelling work highlights important discrepancies with the current work. We have amended the manuscript to include reference to this work now:

"...Further optogenetic labelling work suggests that increased suggests that increased DRN 5-HT firing promotes aspects of reward processing⁸⁵⁻⁸⁷, potentially facilitated by co-release of glutamate and subsequent activation of mesoaccumbens dopamine neurons⁸⁸. Given the methodological disparity between this literature and the present work, drawing direct parallels is challenging, particularly as associations between neuronal firing patterns and synaptic serotonin are region-specific⁸⁹⁻⁹³. "

1.3 Remarks to the Author: *"2. Incidentally, the omission of Rygula et al on the methodological grounds stated in the rebuttal appears curious and illogical. Optogenetic manipulation of 5-HT neurons will almost certainly also produce network/downstream*

effects in areas these neurons project through. Optogenetic manipulations have not so far to my knowledge reproduced effects of regionally and neurochemically selective effects of serotonin depletion, especially in primates. To attribute the selective effects shown by reinforcement theory modelling in that study to "general cognitive impairments" sounds like hand-waving. In fact, the results of Rygula et al support the general view of the optogenetics literature that 5-HT is involved in both reward and punishment processing, (specifically in its anticipation), apparently contradicting the present report. I think the authors should concede that their report will be somewhat controversial in relation to the preclinical literature."

1.3 Author Response: Thank you for this comment. Per your helpful suggestions during the last round of revision, we added a paragraph to highlight the importance of interpreting the findings within the context of its methodological fit to past literature. Rygula *et al.* report that 5-HT depletion reduces sensitivity to misleading reward and punishment during reversal learning, while in the current study participants learn without reversals about explicit reward and punishment outcomes. We have now modified the manuscript to include mention of Rygula *et al.* :

"...consequentially, model-free learning implemented in the present work is challenging to compare with TRP/SSRI work involving model-based or reversal learning¹³⁹⁻¹⁴¹. For example, previous work suggests 5-HT depletion (via pharmacological lesioning) modulates reinforcement sensitivity to misleading punishments and rewards during reversal learning¹⁴²."

1.4 Remarks to the Author: *"3. Minor. The revision on lines 95-97 is missing some words."*

1.4 Author Response: Thank you for this comment, we have revised this sentence in the updated manuscript now.

Responses to Reviewer #2

2.1 Remarks to the Author: *"The revisions of the text and the additional analyses as well as the greater clarity of the analytical (baseline-correction and computational learning modeling) procedures were much appreciated. It is clear the authors have exerted great effort and caution to now fully report all the analyses, data, simulations and parameter recovery. I am now convinced that the various choices are justified."*

2.1 Author Response: Thank you once again for revisiting the manuscript and your positive feedback, this is much appreciated by the authors.

2.2 Remarks to the Author: *"Typo Fig 2 legend: per of symbols should be pair of symbols. Fig 2F one 'in the' too many"*

2.2 Author Response: Thank you for noticing this; we have rectified this typo in the amended manuscript now.

Responses to Reviewer #3

3.1 Remarks to the Author: *"The authors successfully addressed my concerns."*

3.1 Author Response: We would like to thank the reviewer for their feedback, and we are happy to have satisfactorily addressed their concerns.